


**Characterizations and source analysis of atmospheric inorganic ions**
**at a national background site in the northeastern Qinghai-Tibet**
**Plateau: insights into the influence of anthropogenic emissions on a**
**high-altitude area of China**
Bin Han[1], Jing Wang[1], Xueyan Zhao[1], Baohui Yin[1], Xinhua Wang[1], Xiaoyan Dou[2],
Wen Yang[1], Zhipeng Bai[1]
1. State Key Laboratory of Environmental Criteria and Risk Assessment, Chinese
Research Academy of Environmental Sciences, Beijing, China
2. Qinghai Environmental Monitoring Center, Xining, Qinghai, China
**Abstract**
Atmospheric particulate matter (PM) imposes highly uncertain impacts on both
radiative forcing and human health. While ambient PM has been comprehensively
characterized in China's megacities; its composition, source, and characteristics in the
Qinghai-Tibet Plateau (QTP) are not yet fully understood. An autumn observational
campaign was conducted during the 1[st] - 15[th] October 2013 at a national background
monitoring station (3295 m a.s.l.) in the QTP. Real time concentrations of inorganic
water-soluble ions (WSIs) associated with $PM_{2.5}$ were measured in addition to $PM_{2.5}$
concentrations, gaseous pollutants, and meteorological parameters. $SO_4^{2-}$ was the
most abundant WSI (10.00 ±4.39 μg/m$^3$) followed by $NH_4^+$ (2.02 ±0.93 μg/m$^3$), and
$NO_3^-$ (1.65 ±0.71 μg/m$^3$). Observed WSI concentrations were lower as compared to
urban sites in eastern China; however, they were higher as compared to other QTP
monitoring sites. High sulfate and nitrate oxidation ratios indicated strong secondary
formation of both $SO_4^{2-}$ and $NO_3^-$. Both photochemical and heterogeneous reactions
contributed to the formation of particulate $SO_4^{2-}$, while the conversion of $NO_2$ to $NO_3^-$
only occurred via photochemical reactions in the presence of high $O_3$ concentrations
and strong sunlight. Correlation analysis between WSIs revealed that $NH_4NO_3$,
$(NH_4)_2SO_4$, $Na_2SO_4$, and $K_2SO_4$ were the major atmospheric aerosol components. To
better understand the potential sources of WSIs in the QTP, a Positive Matrix
Factorization receptor model was used. Results showed that salt lake emissions,
mixed factor emissions (livestock feces emission, occasional biomass burning, and
crustal material), traffic emissions, secondary inorganic aerosols, and residential
burning were the major emission sources at the study site.


## 1. Introduction

Atmospheric aerosol has a significant impact on climate change and human health, the extent of which is determined by their physical and chemical properties. High concentrations of aerosols are associated with rapid economic growth, urbanization, industrialization, and motorization, and have become a major environmental concern in China (Du et al., 2015). Extensive research has investigated the sources, chemical and physical properties, and evolution processes of aerosol particles at urban and rural sites in China during the last decade (Cao et al., 2007; Gong et al., 2012; He et al., 2011; Jiang et al., 2015; Sun et al., 2015; Sun et al., 2013; Wu et al., 2007b). These studies indicated that fine particles are mainly composed of organics, sulfate, nitrate, ammonium, mineral dust, and black carbon. While these studies have greatly improved our understanding on the sources and physical/chemical properties of aerosol particles, they were predominantly conducted in developed areas of China, including Beijing–Tianjin–Hebei, the Pearl River Delta, and the Yangtze River Delta. In remote areas, such as the Qinghai-Tibet Plateau (QTP), studies on atmospheric aerosol properties are rare.

The QTP covers most of the Tibet Autonomous Region and Qinghai Province in western China, with an area of 5,000,000 km$^2$ and an average elevation over 4000 m. The area is geomorphologically the largest and highest mountain region on earth (Yao et al., 2012). Described as the "water tower" of Asia, this area contains the headwaters of the Mekong, Yangtze, and Yellow Rivers. Therefore, climate variability and change in this region has fundamental impacts on a range of climate-related ecosystem services (McGregor, 2016). Due to its unique ecosystem, landforms, and monsoon circulation, the QTP has a profound role in regional and global atmospheric circulation, radiative budgets, and climate systems (Su et al., 2013; Kopacz et al., 2011; Yang et al., 2014; Jin et al., 2005). Limited anthropogenic activity, a sparse population, immense area, and high elevation mean that, alongside the Arctic and Antarctic, the QTP is considered one of the most pristine terrestrial regions in the world. Because of this, the region is an ideal location for characterizing background aerosol properties, regional and global radiative forcing, climate and ecological changes, and the transportation of global air pollutants. Thus, a comprehensive understanding of QTP aerosol chemistry is crucial for assessing anthropogenic influences and evaluating long-term changes in the global environment (Cong et al., 2015; Zhang et al., 2012).

Research relating to the chemical and physical characteristics of aerosols in the QTP is rare; hence, their sources, properties, and evolution processes are poorly understood. This lack of research is a result of the region's remoteness and challenging weather conditions. Most previous studies of aerosol chemistry in the QTP were conducted in the Himalaya (the southeastern or southern areas of the QTP) to assess the key roles of the Himalaya on regional climate and the environment, as well as the boundary transportation of air pollutants from South Asia (Cong et al., 2015; Zhao et al., 2013b; Wan et al., 2015; Shen et al., 2015). Conversely, the northeastern QTP, located in inland China, is likely to have very different atmospheric


behaviors as compared to those of the Himalaya due to different climate patterns and
aerosol sources between the two regions (Xu et al., 2015). Several studies in the
northeastern QTP found that $SO_4^{2-}$, $NO_3^-$, $NH_4^+$, and $Ca^{2+}$ were major water-soluble
ions (WSIs), suggesting that both anthropogenic pollution and mineral dust
contributed to the total mass of $PM_{2.5}$ (Xu et al., 2014; Li et al., 2013; Zhang et al.,
2014). Du et al. (2015) also found that sulfate and ammonium were dominant in $PM_1$
mass in this area. Other studies from the Waliguan Observatory (36°17'N, 100°54'E,
3816 m a.s.l.), a land-based Global Atmosphere Watch (GAW) baseline station,
located in the northeast of the Tibetan Plateau, found that particles at this site were
predominantly from natural sources, such as soil and crust (Wen et al., 2001; Gao and
Anderson, 2001). However, perturbations from human sources also exist, indicated by
black carbon (BC) concentrations observed at this site (Tang et al., 1999).
Given the rare researches and data on aerosol chemical compositions in the QTP,
more observational data are needed to better characterize the chemical composition of
aerosols in the QTP. WSIs comprise a large portion of aerosol particles and may help
understand chemical reactions in the atmosphere (Tripathee et al., 2017). They can
provide important information for understanding chemical characterizations, sources,
behaviors, and formation mechanisms; and hence, knowledge on the emission of
gaseous precursors and the effect of regional and local pollution on ecosystem health
(Wang et al., 2005; Tripathee et al., 2016). Furthermore, WSIs regulate the electrical
properties of the atmospheric medium, participate in ion-catalyzed and ion–molecule
reactions, and contribute to physicochemical interactions, including ion-induced new
particle formation (Frege et al., 2017; Schulte and Arnold, 1990).
In this study, a real time monitor for WSIs associated with $PM_{2.5}$ was deployed at a
national background monitoring site (Menyuan, Qinghai, 37º36'30"N, 101º15'26"E;
3295 m a.s.l.) in the northeastern QTP, following Du et al. (2015). Hourly mass
concentrations of $PM_{2.5}$ bound sulfate, nitrate, ammonium, sodium, potassium,
magnesium, and calcium were obtained during the 1st –15th October 2013. Real time
measurements of $SO_2$, $NO_X$, CO, $O_3$, $PM_{2.5}$, and meteorological parameters were also
recorded. We discuss the characterization and variation of WSIs; analyze the potential
formation mechanisms of particulate $SO_4^{2-}$ and $NO_3^-$, and investigate potential aerosol
sources by combining WSI and gas pollutant data.

## 2. Methods

### 2.1 Monitoring site

Figure 1 shows the location of the monitoring site at the peak of Daban Mountain,
Menyuan Hui Autonomous County, Qinghai Province (37º36'30"N, 101º15'26"E;
3295 m a.s.l.). The site is owned by the Chinese national atmospheric background
monitoring station system and is approximately 160 km north of Xining, the capital
city of Qinghai Province. The area is characterized by a typical plateau continental
climate with an annual temperature of 0.8 ºC and precipitation of 520 mm.
Meteorological parameters during the observation period are summarized in Table 1.
The site is surrounded by typical QTP vegetation, including potentilla fruticosa and



kobresia. No strong anthropogenic emission sources exist in the adjacent area, with
the exception of occasional biomass burning events and yak dung burning for
residential cooking and heating. The traffic volume around the site is small (Du et al.,
124 2015).

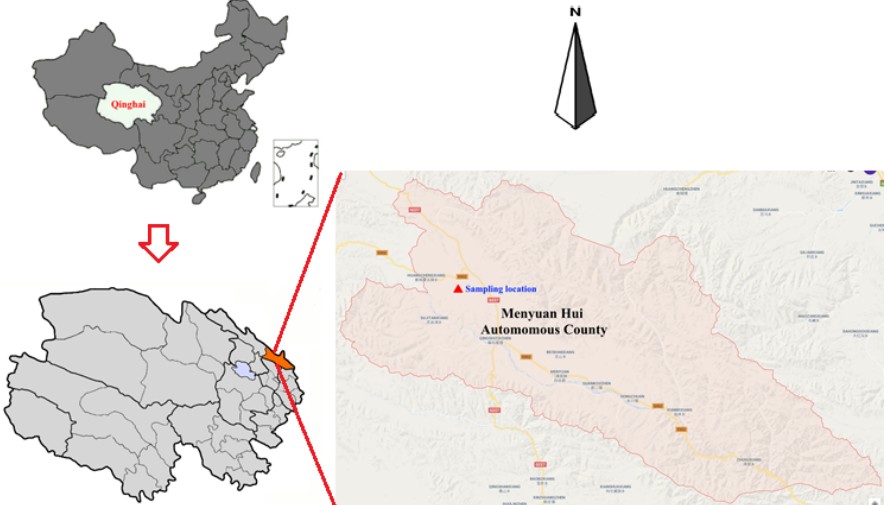

Figure 1 Location of sampling site

**2.2 Instruments**
Hourly concentrations of $NO_3^-$, $SO_4^{2-}$, $Na^+$, $NH_4^+$, $K^+$, $Mg^{2+}$, and $Ca^{2+}$ associated
with $PM_{2.5}$ were simultaneously measured by an ambient ion monitor (Model URG
9000B, URG Corporation, USA). A set of commercial instruments from Teledyne
API (USA) were equipped to measure hourly concentrations of $SO_2$ (M100EU),
$NO/NO_2/NO_x$ (M200EU), CO (M300EU), and $O_3$ (M400E). Hourly $PM_{2.5}$
concentrations were measured using an Ambient Dust Monitor 365 (GRIMM; Grimm
Aerosol Technik GmbH &Co. KG, Ainring, Germany). Meteorological parameters
(e.g. temperature, relative humidity, pressure, and wind speed and direction) were also
recorded.

**2.3 data analysis**
2.3.1 Oxidant ratio
Particulate sulfate and nitrate oxidation ratios (SOR and NOR, respectively),
defined as the molar ratio of $SO_4^{2-}$ and $NO_3^-$ to total oxidized sulfur and nitrogen
(Zhou et al., 2009), were used to evaluate secondary conversion from $NO_2$ and $SO_2$ to
$NO_3^-$ and $SO_4^{2-}$, respectively. High SOR and NOR indicate larger conversions of $SO_2$
and $NO_x$ to their respective particulate forms in $PM_{2.5}$. In this study, NOR and SOR
were calculated based on the following formulae:
$$SOR = \frac{[SO_4^{2-}]}{[SO_2] + [SO_4^{2-}]} \quad (1)$$



$$NOR = \frac{[NO_3^-]}{[NO_2] + [NO_3^-]} \quad (2)$$


2.3.2 Ion balance
Ion balance was used to evaluate the acid-base balance of aerosol particles. We
converted the WSIs mass concentration into an equivalent concentration, as follows:
$$C \,(cation, \mu eq/m^3) = \frac{Na^+}{23} + \frac{NH_4^+}{18} + \frac{K^+}{39} + \frac{Mg^{2+}}{12} + \frac{Ca^{2+}}{20} \quad (3)$$

$$A \,(anion, \mu eq/m^3) = \frac{SO_4^{2-}}{48} + \frac{NO_3^-}{62} \quad (4)$$


2.3.3 Source apportionment
Positive Matrix Factorization (PMF), developed by Paatero (Paatero and Tapper,
1994; Paatero, 1997), has been widely applied in source apportionment researches. In
this model, a data matrix $X_{ij}$, in which $i$ is the sample and $j$ is the measured chemical
species, can be viewed as a speciated data set, and the concept of this model can be
represented as:
$$X_{ij} = \sum_{k=1}^{p} g_{ik} f_{kj} + e_{ij} \quad (5)$$

where $p$ is the number of factors; $f$ is the chemical profile of each source, $g$ is
the mass contribution of each factor to the sample; $f_{jk}$ is the source profile, and $e_{ij}$
is the residual for each species or sample.
PMF solves Eq (5) by minimizing the sum of the square of residuals weighted
inversely with the error estimates of the data points, Q, defined as:
$$Q = \sum_{i=1}^{n} \sum_{j=1}^{m} \left[ \frac{x_{ij} - \sum_{k=1}^{p} g_{ik} f_{kj}}{u_{ij}} \right]^2 \quad (6)$$

where $u_{ij}$ is the uncertainty of chemical species $j$ in sample $i$.
Uncertainty for each species can be calculated using following equation:
$$\mathrm{Unc} = \sqrt{(Error\ Fraction \times concentration)^2 + (MDL)^2} \quad (7)$$

where MDL is the method detective limit of each species.
Given that PMF is a descriptive model, there are no objective criteria for
choosing the appropriate number of factors (Paatero et al., 2002). Therefore, several
criteria were applied, including extracting realistic source profiles, distribution of
scaled residuals, Q/Qexp, and the comparison between the PMF modeled and
measured elemental mass (Crilley et al., 2017). In this study, the Q/Q$_{exp}$ index was
monitored with an increasing number of factors (3–6), because a large decrease is
indicative of increased explanatory power, while a small decrease is suggestive of
little improvement with additional factors. Consequently, the number of factors was
chosen after Q/Q$_{exp}$ decreased significantly. Q$_{exp}$ was then calculated using the
following equation:
$$Q_{exp} = N_{sample} \times M_{good} + \frac{N_{sample} \times M_{weak}}{3} - (N_{sample} \times P_{factor}) \quad (8)$$



where $N_{sample}$ is the number of samples in the model; $M_{good}$ and $M_{weak}$ are
the number of good or weak model species, respectively; and $P_{factor}$ is the number
of estimated factors.
An argument on the application of PMF in this study is that the number of
components associated with $PM_{2.5}$ is limited; therefore, more available data should be
introduced into the model to ensure better source information and model results
(Hopke, 2010). In previous studies, a practical way to extract more source information
from the available data was to include data of other air pollutants such as volatile
organic compounds (VOCs) (Wu et al., 2007a; Mo et al., 2017), major gaseous
pollutants ($NO_x$, $SO_2$, and CO) (Rizzo and Scheff, 2007; Masiol et al., 2017), particle
size distribution data (Zhou et al., 2005), and air trajectory and meteorological data
(Buzcu-Guven et al., 2007). In this study, we introduced gaseous pollutants data,
combined them with WSIs data, and applied the new dataset into the PMF model.

2.3.4 Statistical analysis
Correlation analysis, analysis of variation (ANOVA), and linear regression were
applied. All statistical calculations were preformed using R studio software packages
(Version 0.99.903, RStudio, Inc.).

**3. Results and Discussion**
**3.1 Descriptive analysis**
Table 1 summarizes the concentrations of WSIs, $PM_{2.5}$, and gaseous pollutants
and data of meteorological parameters during the observation period. $SO_4^{2-}$ accounted
for 67.9% of the total WSIs mass, followed by $NH_4^+$ (13.7%), and $NO_3^-$ (11.2%).
$SO_4^{2-}$, $NO_3^-$ and $NH_4^+$ (SNA), accounting for 92.8% of the total WSIs mass, were the
major components of secondary inorganic aerosols.





Table 1 Descriptive statistics of WSIs species, gaseous pollutants and meteorological parameters

| Species | Mean | Standard Deviation | Percentile | | | | |
|---|---|---|---|---|---|---|---|
| | | | 5th | 25th | 50th | 75th | 95th |
| WSIs (µg/m³) | | | | | | | |
| $NO_3^-$ | 1.65 | 0.71 | 0.62 | 1.14 | 1.60 | 2.02 | 2.90 |
| $SO_4^{2-}$ | 10.00 | 4.39 | 6.39 | 7.05 | 8.37 | 10.73 | 18.83 |
| $Na^+$ | 0.86 | 0.61 | 0.01 | 0.50 | 0.79 | 1.12 | 1.66 |
| $NH_4^+$ | 2.02 | 0.93 | 0.52 | 1.40 | 1.95 | 2.54 | 3.59 |
| $K^+$ | 0.05 | 0.03 | 0.01 | 0.03 | 0.04 | 0.06 | 0.11 |
| $Mg^{2+}$ | 0.06 | 0.19 | 0.01 | 0.02 | 0.04 | 0.05 | 0.09 |
| $Ca^{2+}$ | 0.09 | 0.05 | 0.01 | 0.05 | 0.09 | 0.11 | 0.18 |
| $[NO_3^-]/[SO_4^{2-}]$ | 0.29 | 0.13 | 0.11 | 0.19 | 0.29 | 0.37 | 0.49 |
| Air pollutants (µg/m³) | | | | | | | |
| $PM_{2.5}$ | 18.99 | 13.10 | 2.60 | 9.00 | 16.15 | 26.35 | 44.28 |
| $SO_2$ | 4.37 | 5.76 | 1.28 | 1.79 | 2.40 | 3.88 | 14.70 |
| NO | 0.12 | 0.19 | 0.01 | 0.02 | 0.04 | 0.13 | 0.45 |
| $NO_2$ | 4.35 | 2.66 | 1.42 | 2.58 | 3.96 | 5.26 | 8.99 |
| $NO_X$ | 4.45 | 2.70 | 1.42 | 2.68 | 4.14 | 5.33 | 9.02 |
| CO | 48.59 | 56.51 | 4.60 | 13.65 | 26.31 | 58.35 | 183.57 |
| O3 | 107.71 | 25.13 | 82.91 | 92.74 | 106.93 | 117.72 | 134.02 |
| Acidity (µeq/m³) | | | | | | | |
| Anion | 0.22 | 0.11 | 0.01 | 0.17 | 0.19 | 0.26 | 0.42 |
| Cation | 0.13 | 0.08 | 0.00 | 0.05 | 0.14 | 0.17 | 0.25 |
| Oxidation Ratio | | | | | | | |
| SOR | 0.65 | 0.16 | 0.34 | 0.56 | 0.69 | 0.75 | 0.88 |
| NOR | 0.22 | 0.10 | 0.10 | 0.16 | 0.20 | 0.26 | 0.38 |
| Meteorological Parameters | | | | | | | |
| Temp (°C) | 6.55 | 4.53 | 0.50 | 2.80 | 5.60 | 10.70 | 14.49 |
| RH (%) | 52.92 | 18.44 | 22.56 | 38.40 | 55.80 | 65.80 | 81.56 |
| Pressure (kPa) | 68.58 | 0.26 | 68.10 | 68.30 | 68.60 | 68.80 | 68.90 |
| Wind Speed (m/s) | 3.39 | 3.02 | 0.00 | 1.30 | 2.60 | 4.70 | 9.20 |



Table 2 Comparisons of WSIs concentrations with other high altitude and urban sites (mean, μg/m³)

| Sampling site | Sampling Year | $SO_4^{2-}$ | $NO_3^-$ | $Na^+$ | $NH_4^+$ | $K^+$ | $Ca^{2+}$ | $Mg^{2+}$ | $[NO_3^-]/[SO_4^{2-}]$ | Reference |
|---|---|---|---|---|---|---|---|---|---|---|
| The QTP site | | | | | | | | | | |
| Menyuan, Qinghan, the northeastern QTP (3295m) | 2013 | 10.0 | 1.6 | 0.9 | 2.09 | 0.05 | 0.09 | 0.06 | 0.29 | This study |
| South edge of the QTP (4276m) | 2009 | 0.43 | 0.20 | 0.07 | 0.03 | 0.02 | 0.88 | 0.04 | 0.72 | Cong et al. (2015) |
| Qilian Shan Station, the northeastern QTP (4180m) | 2010 | 0.74 | 0.20 | 0.04 | 0.15 | 0.04 | 0.18 | 0.04 | 0.42 | Xu et al. (2014) |
| Qinghai Lake, the northeastern QTP (3200m) | 2010 | 4.45 | 0.38 | 0.13 | - | 0.12 | 0.23 | 0.06 | 0.13 | Zhang et al. (2014) |
| | 2012 | 3.65 | 1.42 | 0.26 | 0.62 | 0.10 | 0.66 | 0.10 | 0.60 | Zhao et al. (2015) |
| Low altitude site in China (urban and background sites) | | | | | | | | | | |
| Sampling site | Sampling Year | $SO_4^{2-}$ | $NO_3^-$ | $Na^+$ | $NH_4^+$ | $K^+$ | $Ca^{2+}$ | $Mg^{2+}$ | $[NO_3^-]/[SO_4^{2-}]$ | Reference |
| Beijing (43m) | 2013 | 13.80 | 15.43 | 0.69 | 8.02 | 1.06 | 0.08 | 0.58 | 1.28 | Dao et al. (2014) |
| Shanghai (4m) | 2009 | 12.9 | 15.0 | - | 6.64 | 0.94 | - | - | 1.80 | Ming et al. (2017) |
| Xi'an (396m) | 2006 | 42.0 | 20.6 | - | 13.1 | - | - | - | 0.76 | Zhang et al. (2011) |
| Chongqing (160m) | 2012 | 15.4 | 8.4 | 0.25 | 6.9 | 0.67 | 0.24 | 0.04 | 0.84 | Chen et al. (2017) |
| Shangdianzi, Beijing (293m) | 2009 | 8.68 | 11.20 | - | 3.23 | - | - | - | 2.00 | Zhao et al. (2013a) |
| Lin'an, Zhejiang (138m) | 2014 | 15.9 | 11.7 | 2.6 | 4.9 | 1.1 | 3.7 | 0.2 | 1.14 | Zhang et al. (2017) |
| Other high-altitude area sites around the world | | | | | | | | | | |
| Sampling site | Sampling Year | $SO_4^{2-}$ | $NO_3^-$ | $Na^+$ | $NH_4^+$ | $K^+$ | $Ca^{2+}$ | $Mg^{2+}$ | $[NO_3^-]/[SO_4^{2-}]$ | Reference |
| Langtang, remote Himalayas, Nepal (3920m) | 1999-2000 | 0.27 | 0.04 | 0.06 | 0.15 | 0.02 | 0.03 | 0.004 | 0.23 | (Carrico et al., 2003) |
| Nagarkot, Kathmandu Valley, Nepal (2150m) | 1999-2000 | 2.5 | 0.8 | 0.13 | 1.2 | 0.28 | 0.05 | 0.01 | 0.50 | (Carrico et al., 2003) |
| Gurushikhar in Mt. Abu, Indian (1680 m) | 2007 | 3.56 | - | 0.28 | 0.92 | 0.10 | 0.19 | 0.06 | - | (Kumar and Sarin, 2010) |
| Golden, Colorado, USA (1850 m) | 2014 | 0.67 | 0.40 | - | 0.60 | - | - | - | 0.92 | (Valerino et al., 2017) |
| Monte Martano, Italy (1100 m) | 2009 | 1.90 | 0.84 | 0.02 | 0.54 | 0.06 | 0.25 | 0.06 | 0.68 | (Moroni et al., 2015) |
| Lassen Volcanic National Park, California, USA (1798 | 2009-2012 | 0.35 | 0.12 | - | - | - | 0.04 | - | 0.53 | (VanCuren and Gustin, 2015) |





| | m) | | | | | | | | | |
|---|---|---|---|---|---|---|---|---|---|---|
| Great Basin National Park, Nevada, USA (2060 m) | 2009-2012 | 0.38 | 0.10 | - | - | - | 0.05 | - | 0.41 | (VanCuren and Gustin, 2015) |

To better understand the concentrations of WSIs, we compared our observations
with other studies implemented in background sites or urban sites across China and
high altitude areas around the world in Table 2. Our results are lower as compared to
studies in Europe and the USA (VanCuren and Gustin, 2015; Valerino et al., 2017;
Moroni et al., 2015), and the high latitude Himalaya region (Carrico et al., 2003);
however, observations are comparable with some urban area in India and Nepal
(Carrico et al., 2003; Kumar and Sarin, 2010). Observed concentrations of $SO_4^{2-}$ are
also lower as compared to low altitude sites in China, for example urban sites in
Beijing, Shanghai, Xi'an, and Chongqing and background sites in Shangdianzi
(Beijing) and Lin'an (Zhejiang). Concentrations of $NO_3^-$ were five to thirteen times
lower as compared to those in low altitude areas (both urban and background sites),
indicating that the influence of vehicle emissions in studying area is weak. $NH_4^+$
levels were lower as compared to those in urban sites (three to six times lower), and
also slightly lower as compared to background sites (less than three times lower).
SNA concentrations in this study were higher as compared to those at other sites
in the QTP, including the southern edge (Cong et al., 2015), Qilian Shan Station (Xu
et al., 2014), and Qinghai Lake in the northeastern QTP (Zhang et al., 2014; Zhao et
al., 2015). Large differences in concentrations suggest that the monitoring site in this
study appears to be more impacted by natural and human activities as compared to
other sites in the QTP.
Tables 1 and 2 show the molar ratios of $NO_3^-$ and $SO_4^{2-}$, an indicator of the
relative importance of vehicle versus coal combustion emissions in the atmosphere
(Arimoto et al., 1996; Yao et al., 2002). The ratio in this study (0.29±0.13) is lower as
compared to low altitude areas, which are characterized as vehicle emission dominant
or co-dominant by vehicle and coal emissions. The ratio varies apparently across the
different sites in the QTP as shown in Table 2. Even at a single monitoring site
(Qinghai Lake; Table 2), different studies reported different ratios (Zhang et al., 2014;
Zhao et al., 2015). Thus, it is likely that remote transportation and local emissions
jointly affect air pollution over these monitoring sites.





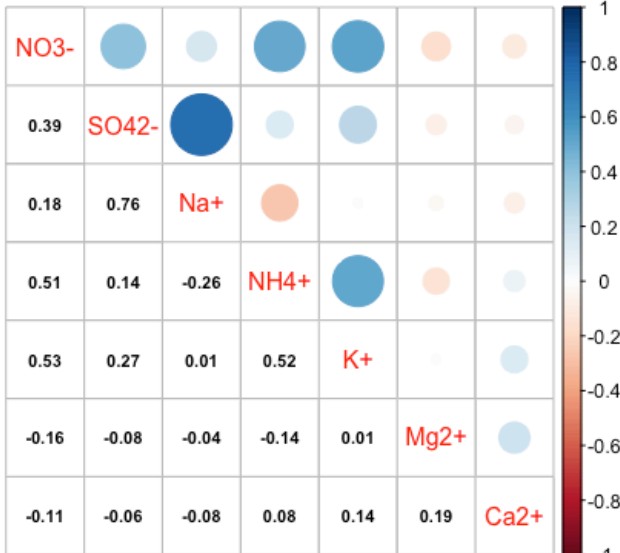


Figure 2 Correlation coefficients (r) between WSIs in PM$_{2.5}$ during sampling period


Correlations between WSIs are useful to investigate potential associations
between the various WSIs (Xu et al., 2014). Figure 2 illustrates the correlation
coefficients between WSIs based on their mass concentrations. A high correlation was
found between Na$^+$ and SO$_4^{2-}$ (r=0.76). NO$_3^-$ and SO$_4^{2-}$ had a negative and weak
correlation with Mg$^{2+}$ and Ca$^{2+}$, which were found to be highly correlated with CO$_3^{2-}$
in another study in the QTP (Xu et al., 2014). SNA displayed medium positive
correlations with each other. K$^+$, commonly used as a marker for emissions from the
burning of biomass or biofuel, had a medium correlation with NH$_4^+$ and NO$_3^-$.
To further examine the relationship between PM$_{2.5}$ and WSIs, we divided the
PM$_{2.5}$ concentrations into four categories: a) C(PM$_{2.5}$) < 20μg/m$^3$, (b) 20μg/m$^3$ ≤
C(PM$_{2.5}$) <30μg/m$^3$, (c) 30μg/m$^3$ ≤ C(PM$_{2.5}$) <40μg/m$^3$, and (d) C(PM$_{2.5}$) ≥ 40μg/m$^3$
and attributed each WSI measurement to its corresponding PM$_{2.5}$ category. Figure 3
shows the mean proportions of WSIs in PM$_{2.5}$ for the different categories. As the
PM$_{2.5}$ concentration increases, the percentages of WSIs in PM$_{2.5}$ mass exhibited
decreasing trends, suggesting that the contribution of WSIs to PM$_{2.5}$ increases was
negligible. Du et al. (2015) deployed an Aerodyne Aerosol Chemical Speciation
Monitor simultaneously with our study, and found that organics were the only species
that increased during the particle growth period. This finding suggests that organics,
rather than WSIs, are playing a dominant role in particle growth at the national
background site, rather than ammonium sulfate. Other studies have also confirmed
this finding (Dusek et al., 2010; Ehn et al., 2014; Setyan et al., 2014).





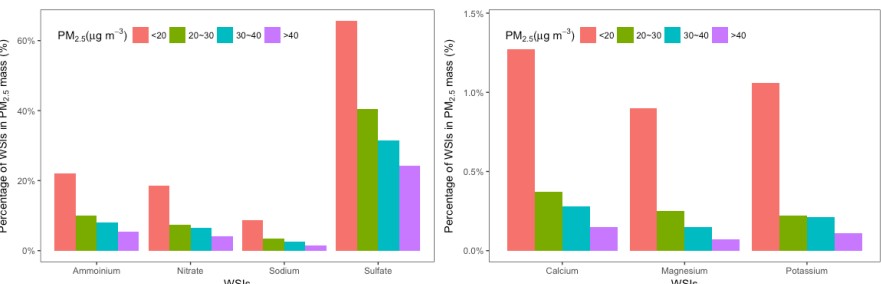


273          Figure 3 Mass portions of WSIs within different PM$_{2.5}$ level ranges


## 3.2 Diurnal variation analysis

Diurnal variations of WSIs in PM$_{2.5}$, related gaseous pollutants (SO$_2$, NO$_2$, and
O$_3$), and meteorological parameters (temperature, relative humidity, and wind speed)
are shown in Figure 4.

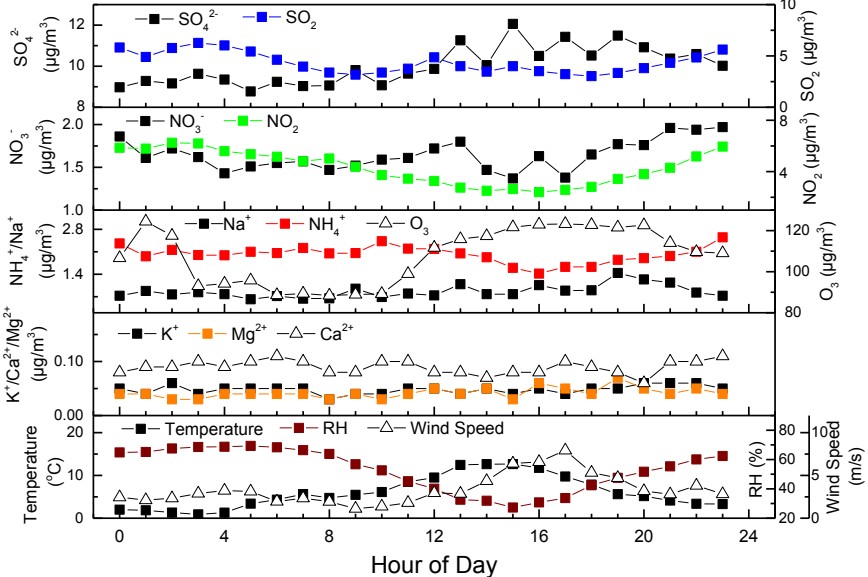


Figure 4 Diurnal variations of WSIs in PM$_{2.5}$, gaseous pollutants (SO$_2$, NO$_2$, O$_3$), as well as

281       meteorological parameters (temperature, relative humidity and wind speed) during sampling

282                                          period

SO$_4^{2-}$ concentrations begin to increase from midnight (Beijing time), reach peak
levels at approximately 15:00, and then decrease gradually. SO$_2$ concentrations
exhibit a bimodal trend, with peaks at 03:00 and 12:00; conversely, SO$_4^{2-}$ exhibits an
inverse trend. NO$_3^-$ concentrations peak at midnight and in the early afternoon, with
lowest levels occurring during the late afternoon. NO$_2$ displays high nighttime levels
and low daytime levels. NH$_4^+$ remains steady during the morning with a peak at 10:00.
O$_3$, temperature, RH, and wind speed also display evident diurnal variations; O$_3$,
temperature, and wind speed are low (high) at night (day), while RH shows an inverse





variation to this pattern.

### 3.3 Sulfate and nitrate oxidation ratio analysis

Average NOR and SOR during the whole measurement campaign were 0.22 and
0.65, respectively, suggesting potentially strong secondary formation of both $SO_4^{2-}$
and $NO_3^-$. Strong photochemical reactions and the existence of high $O_3$ concentrations
would elevate the oxidant ratio from $SO_2$ and $NO_2$ to $SO_4^{2-}$ and $NO_3^-$, despite the low
intensity of local emissions. Furthermore, remote transportation also contributed to
the concentrations of $SO_4^{2-}$ and $NO_3^-$, thus increasing the oxidant ratio, despite not
being products of local oxidation.
Variation trends of SOR and NOR were compared to changes in $PM_{2.5}$
concentration and ambient RH, as shown in Figure 5. Previous research has shown
that, in urban areas, both SOR and NOR increased with the $PM_{2.5}$ concentration,
suggesting that heavy $PM_{2.5}$ pollution corresponds to high SOR and NOR (Xu et al.,
2017). However, studies investigating SOR and NOR variations at background sites
with low $PM_{2.5}$ concentration range are rare. Our results showed that increasing
concentrations of $PM_{2.5}$ at low levels corresponded to decreasing SOR and NOR
(Figure 5a), although the decreases were slight. This is consistent with our previous
finding that concentrations of $SO_4^{2-}$ and $NO_3^-$ did not vary markedly with $PM_{2.5}$
increases at background sites, and provides further evidence to suggest that $SO_4^{2-}$ and
$NO_3^-$ are not key drivers of $PM_{2.5}$ growth at low levels. The simultaneous observations
of Du et al. (2015), indicated that organics were thought to be the major driver of
particle growth.
In Figure 5b, SOR initially decreases and then increases as RH increases. Peak
SOR occurs when RH reaches both its maximum and minimum levels, when RH is
low (10–20%), $O_3$ is high (114.6 μg/m$^3$, approximately the 70$^{th}$ percentile of $O_3$
concentrations), and vice versa (RH > 70% and $O_3$ 93.8 μg/m$^3$, approximately the 30$^{th}$
percentile of $O_3$ concentrations). The formation of particulate $SO_4^{2-}$ can be achieved
via aqueous-phase oxidation (heterogeneous reaction) or gas-phase oxidation
(photochemical reaction). Normally, aqueous-phase oxidation from $SO_2$ to $SO_4^{2-}$ is
faster than gas-phase oxidation (Wang et al., 2016). When RH is low and $O_3$ is high,
the photochemical formation of $SO_4^{2-}$ via gas-phase oxidation should be considered
the main oxidation pathway. Conversely, low $O_3$ and high RH are not sufficient to
provide adequate oxidizing capacity; thus photochemical $SO_4^{2-}$ formation becomes
less important and aqueous-phase oxidation plays a more dominant role.
NOR constantly decreases as RH increases. Particulate $NO_3^-$ is predominantly
formed by the gas-phase reaction of $NO_2$ and OH radicals during the day and by
heterogeneous reactions of nitrate radicals ($NO_3$) at night (Seinfeld and Pandis, 2016).
In this study, high (low) $O_3$ and low (high) RH lead to high (low) NOR, meaning that
gas-phase reactions oxidized by high levels of $O_3$ are the major pathway for nitrate
formation, while heterogeneous reactions play a less important role.
Trends of SOR and NOR with RH and $O_3$ suggest that both photochemical and
heterogeneous reactions contribute to the secondary transformation of $SO_2$, while only
photochemical reaction drives the conversion of $NO_2$ to nitrate.

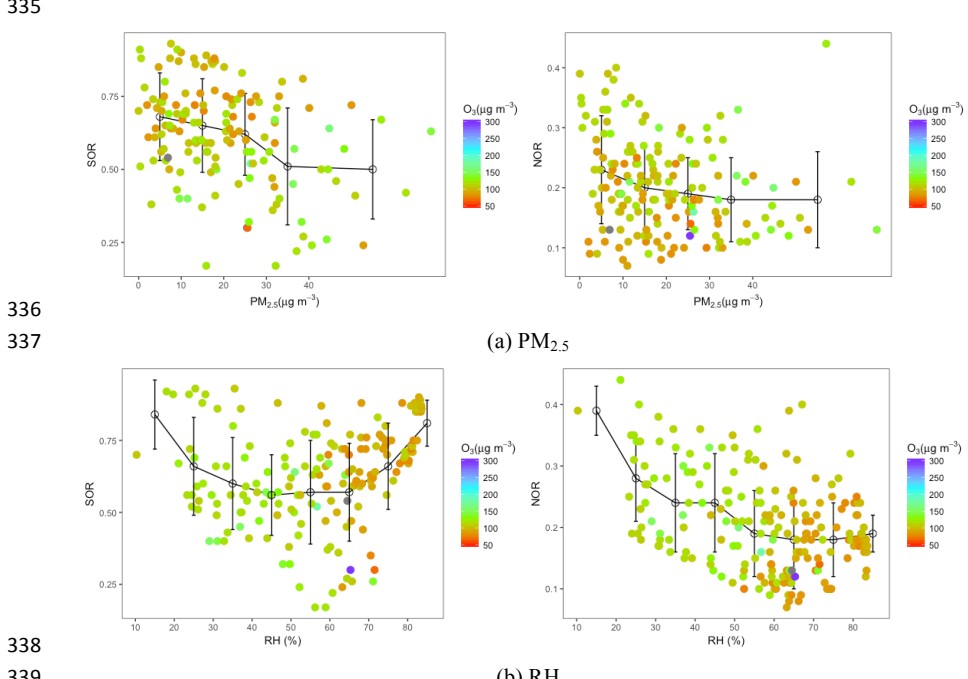

(a) PM$_{2.5}$

(b) RH

Figure 5 Variations of SOR and NOR as a function of PM$_{2.5}$ and RH. The vertical bars correspond to one standard error from the mean.

Figure 6 characterizes the diurnal variations of SO$_4^{2-}$, NO$_3^-$, SOR, NOR, O$_3$, and RH. The variation of SOR is small, particularly as compared to the evident diurnal variation in SO$_4^{2-}$. Daytime gas-phase oxidation and nighttime aqueous-phase oxidation are thought to be equally important to the formation of SO$_4^{2-}$. NOR is high during the day and low at night; reflecting a strong positive correlation with O$_3$ (r=0.71, p<0.05) and a weak negative correlation with RH (r=-0.43, p<0.05). The high correlation between NOR and O$_3$ indicates that gas-phase oxidation via photochemical reactions is the main NO$_3^-$ formation pathway.

It is apparent that photochemical reactions (dominated by O$_3$ oxidization) contribute markedly to the secondary conversion of both SO$_2$ and NO$_2$, while heterogeneous reactions (promoted by the existence of aqueous phase) contributed to the formation of SO$_4^{2-}$ and only had a weak effect on NO$_3^-$. Ma et al. (2003) found that fine nitrate particles (Dp < 2.0 μm) at Waliguan Observatory (150 km south of our monitoring site) were most likely produced via gaseous-phase reactions between nitric acid and ammonia, in line with our findings on nitrate formation.



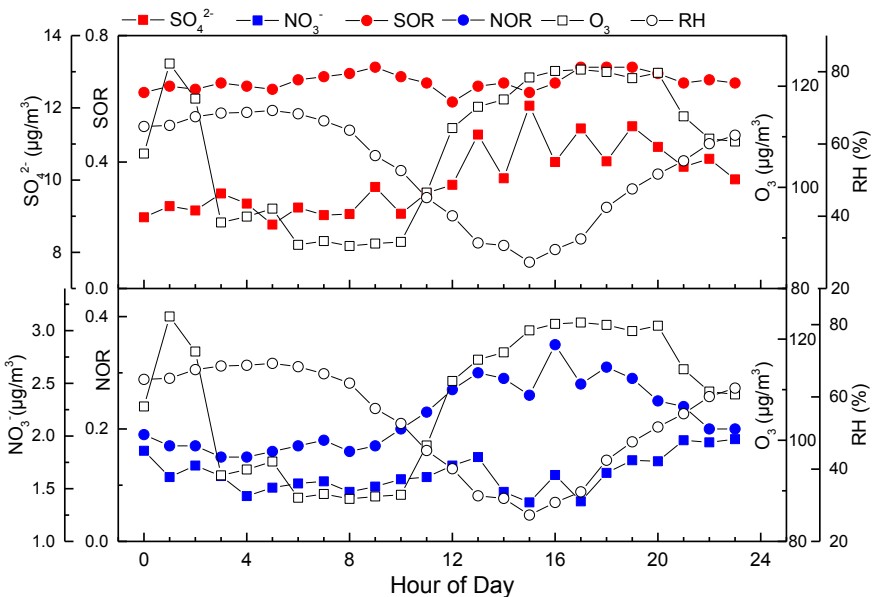


360        Figure 6 Diurnal variations of $SO_4^{2-}$ and $NO_3^-$, SOR, NOR $O_3$ and RH during sampling period


**3.4 Molecular composition of major ionic species**

363        The molecular chemical forms of the major WSIs in $PM_{2.5}$ were identified using
bivariate correlations based on individual WSI molar concentrations (Verma et al.,
2010; Wang et al., 2005). In this study, we used equivalent concentrations for
correlation analysis, and the coefficients are shown in Table 3. Figure 7 a–c show
scatter plots of the equivalent concentrations of $[NH_4^+]$ and $[NO_3^-]$, $[Na^+]$ and $[SO_4^{2-}]$,
and $[NH_4^+]$ and $[SO_4^{2-} + NO_3^-]$, respectively. $(NH_4)_2SO_4$ and $NH_4NO_3$ are major
components of atmospheric aerosols (Park et al., 2004), commonly formed by the
neutralization of sulfuric acid ($H_2SO_4$) and nitric acid ($HNO_3$) by $NH_3$ (Xu et al.,
2014). It is apparent that $NH_4^+$ is closely correlated with $NO_3^-$ (r=0.56). The slope of
the regression between $NH_4^+$ and $NO_3^-$ ($\mu ep/m^3$ versus $\mu ep/m^3$) is 2.28, indicating the
complete neutralization of $NO_3^-$ by $NH_4^+$. $SO_4^{2-}$ was highly correlated with $Na^+$
(r=0.56), rather than $NH_4^+$. Unlike $NO_3^-$ and $NH_4^+$, $Na^+$ was completely neutralized by
$SO_4^{2-}$ (slope=0.15) in the form of $NaHSO_4$. Excess $NH_4^+$ would then combine with
excess $SO_4^{2-}$ (r=0.46). The regression slope between excess $NH_4^+$ and excess $SO_4^{2-}$
was 0.72, meaning that excess $NH_4^+$ was completely neutralized by $SO_4^{2-}$ and existed
in the form of $(NH_4)_2SO_4$. Excess sulfuric acid was likely neutralized by crustal WSIs;
and $K_2SO_4$ was a major chemical species in aerosol particles based on their
correlation coefficients. Good regression results between $[NH_4^+ + Na^+ + K^+]$ and
$[NO_3^- + SO_4^{2-}]$ were also observed in Figure 7 d.





Table 3 Correlation coefficients (r) between the equivalent concentrations of WSIs in PM$_{2.5}$
during sampling period

|  | NO$_3^-$ | SO$_4^{2-}$ | Na$^+$ | NH$_4^+$ | K$^+$ | Mg$^{2+}$ | Ca$^{2+}$ |
|---|---|---|---|---|---|---|---|
| NO$_3^-$ |  |  |  |  |  |  |  |
| SO$_4^{2-}$ | 0.49 |  |  |  |  |  |  |
| Na$^+$ | 0.17 | 0.56 |  |  |  |  |  |
| NH$_4^+$ | 0.56 | 0.46 | -0.24 |  |  |  |  |
| K$^+$ | 0.56 | 0.39 | 0.06 | 0.57 |  |  |  |
| Mg$^{2+}$ | -0.11 | -0.13 | -0.06 | -0.15 | -0.03 |  |  |
| Ca$^{2+}$ | -0.06 | -0.11 | 0.01 | -0.08 | 0.03 | 0.09 |  |


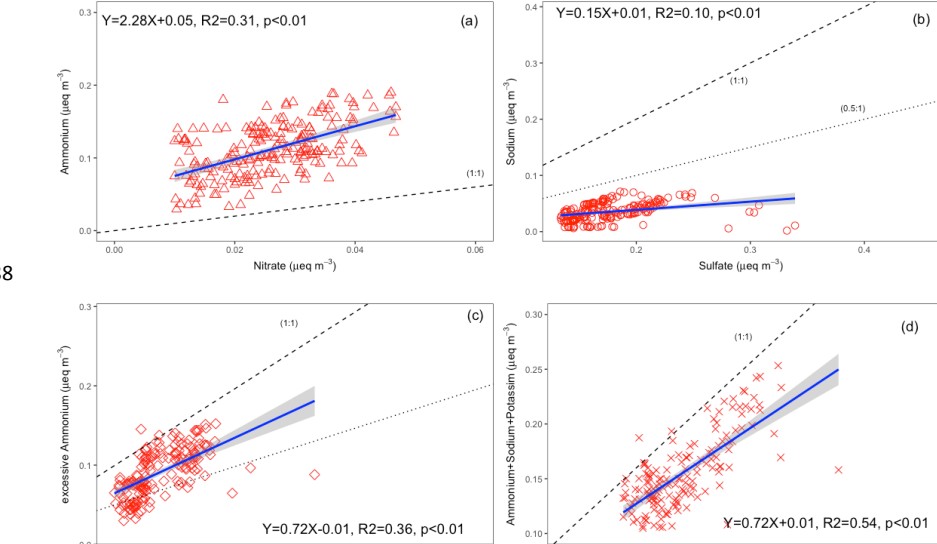



Figure 7 Scatter plot of (a) [NH$_4^+$] and [NO$_3^-$] (b) [Na$^+$] and [SO$_4^{2-}$] (c) [excessive NH$_4^+$] and
[excessive SO$_4^{2-}$] (d) [NH$_4^+$ + Na$^+$ + K$^+$] and [NO$_3^-$ + SO$_4^{2-}$]

**3.5 Ion acidity analysis**
The ion balance, expressed by the sum of the equivalent concentration (μeq/m$^3$)
ratio of cation to anion (C/A), is an indicator of the acidity of particulate matter
(Wang et al., 2005). In this study, the ion balance ratio was 0.87, indicating that
aerosols tended to be acid, in line with previous studies in the QTP (Xu et al., 2015;
Zhao et al., 2015). This suggests that anthropogenic emissions (e.g. SO$_4^{2-}$ and NO$_3^-$),
either regional or local, impacted aerosol acidity during the observation period, and
that the contribution from mineral dust was weak.
Figure 8 shows the scatter and linear regression plot of cations and anions
(μeq/m$^3$). It is apparent that most points are below the 1:1 line, highlighting the acid




tendency. The total equivalent anion concentration was regressed against the total
equivalent concentrations of cations, and the slope of regression was 0.58.

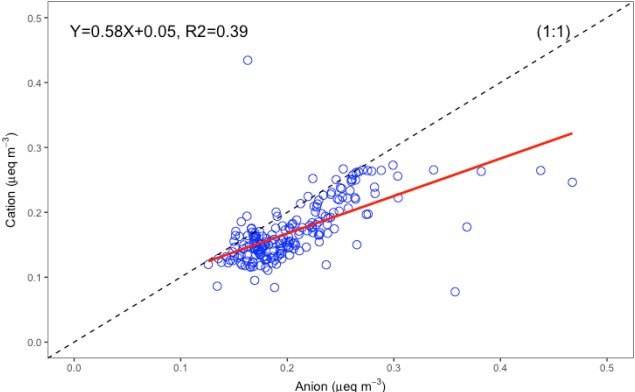


Figure 8 Cation and Anion scatter plot and linear regression

Aerosol acidity for different categories of PM$_{2.5}$ concentration (described in
section 3.1) and their respective scatter plots of total equivalent concentrations of
anions and cations are shown in Figure 9. C/A was high when PM$_{2.5}$ concentrations
exceeded 20μg/m$^3$, indicating that aerosol acidity was weak when the PM$_{2.5}$
concentration was high. This provides further evidence to support our finding that
SO$_4^{2-}$ and NO$_3^-$ did not contribute to PM$_{2.5}$ increases.

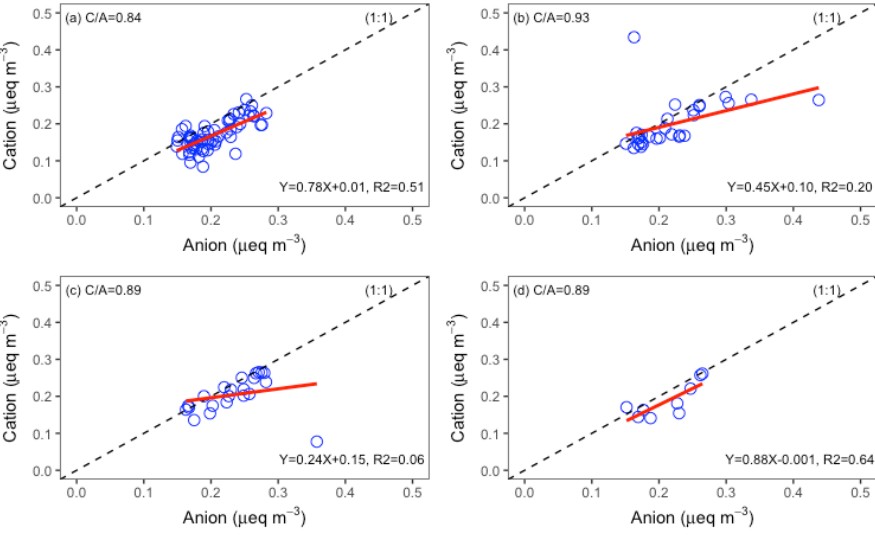


Figure 9 Scatter plot Σ Anion and Σ Cation with different PM$_{2.5}$ concentration range (a) C(PM$_{2.5}$)
< 20μg/m$^3$ (b) 20μg/m$^3$ ≤ C(PM$_{2.5}$) <30μg/m$^3$ (c) 30μg/m$^3$ ≤ C(PM$_{2.5}$) <40μg/m$^3$ (d) C(PM$_{2.5}$) ≥
40μg/m$^3$





### 3.6 Source apportionment by PMF

421  In this study, all WSIs and gaseous pollutants were introduced into the PMF
model for source identification. We ran the PMF model with different numbers of
factors to determine the $Q/Q_{exp}$ variation. The $Q/Q_{exp}$ decrease between four and five
factors was the largest (Figure 10); therefore, five factors were used in the PMF
model. The distributions of the factor species and the percentage of total species are
shown in Figure 11.

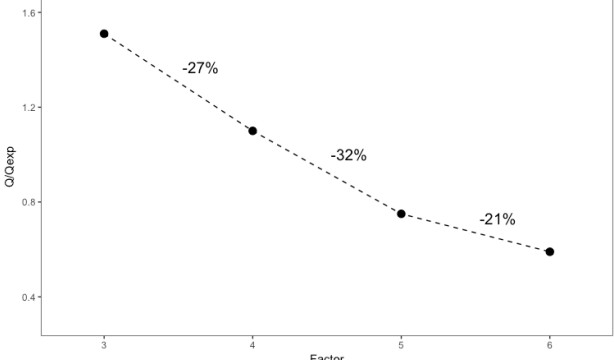


429   Figure 10 The decrease ratio of $Q/Q_{exp}$ with different number of factor choice






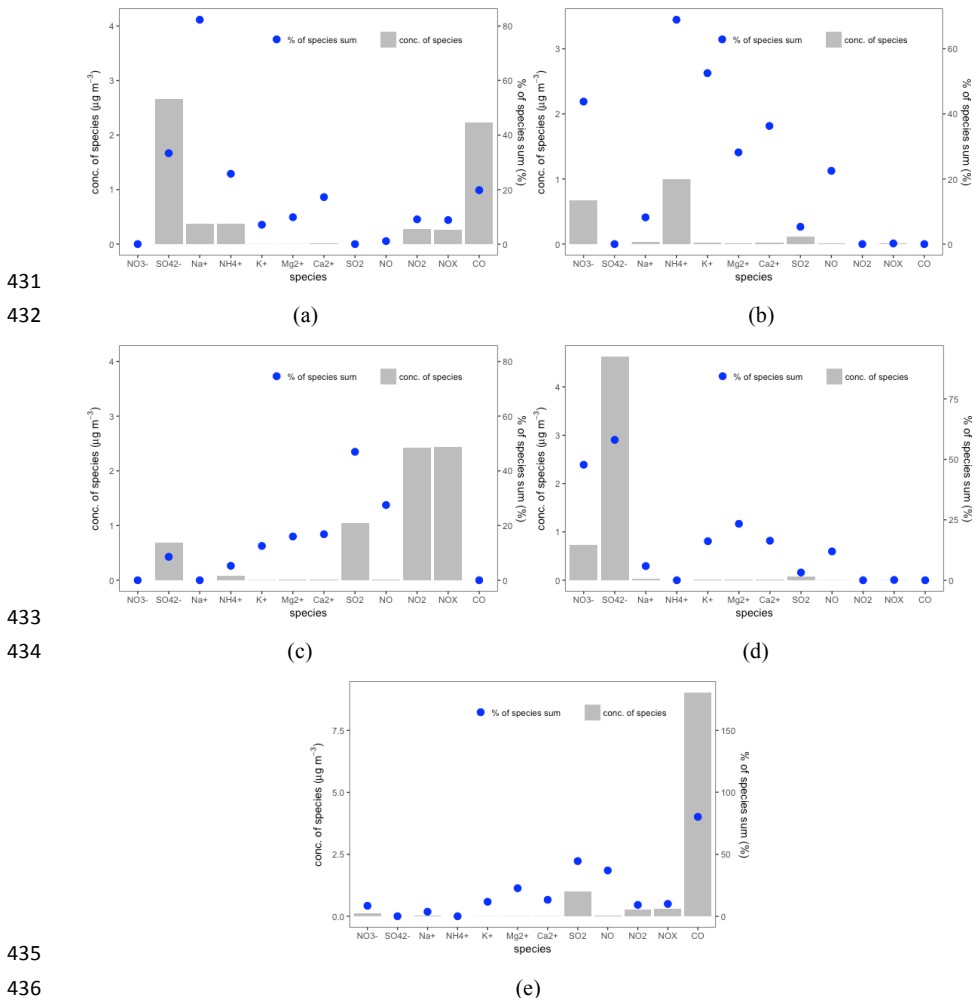


(a) (b)

(c) (d)

(e)
Figure 11 Source profiles exacted by PMF (a) Factor 1: salt lake emissions; (b) Factor 2: Crustal
dust, animal waste emission and biomass burning;   (c) Factor 3: Vehicle emissions; (d) Factor 4:
Secondary inorganic aerosol; (e) Domestic burning (black circle represented the percentage of
species sum, while white bar represented concentration of species)

Factor 1 has high $Na^+$ loading and a moderate $SO_4^{2-}$ loading. Our monitoring site
is approximately 100 km from Qinghai Lake, a saline and alkaline water body which
is the largest lake in China. Zhang et al. (2014) collected total suspended particle
(TSP) and $PM_{2.5}$ samples at Qinghai Lake, and found that the concentration of $Na^+$
was higher as compared to other mountainous areas. Furthermore, they found that
$SO_4^{2-}$ was one the most abundant species in both TSP and $PM_{2.5}$. Therefore, we
attribute factor 1 to aerosols emitted from Qinghai Lake.
High $NH_4^+$ and $K^+$ loading and moderate $Mg^{2+}$ and $Ca^{2+}$ loading were observed in
factor 2. Livestock feces, which is commonly found in the meadows around the



sampling site is a possible source of $NH_4^+$. $K^+$ can be attributed to the combustion of biomass, which has both natural and anthropogenic origins. Li et al. (2015) found that occasional biomass burning events in the area contributed significantly to the formation of anthropogenic fine particles. Moreover, crustal materials were considered as major sources of $Mg^{2+}$ and $Ca^{2+}$ (Ma et al., 2003). Consequently, factor 2 is attributed to mixed sources, including livestock waste emissions, biomass burning, and crustal materials.

Factor 3 is identified as traffic emissions due to high contributions of $NO_2$ and $NO_x$, and a moderate $SO_2$ loading. A national road (G227) and a provincial road (S302) pass near the monitoring site, and many of the transient vehicles are heavy-duty trucks. Consequently, road emissions do influence the monitoring site, irrespective of the low traffic volume.

Factor 4 is secondary inorganic aerosol, enriched with $SO_4^{2-}$ and $NO_3^-$. The precursor of $SO_4^{2-}$ is $SO_2$, which may originate from coal combustion, and $NO_3^-$ is mainly converted from ambient $NO_x$, emitted by both vehicle exhaust and fossil fuel combustion.

Factor 5 has a CO high loading and moderate $SO_2$ and NO loading. Residential cooking and heating often causes heavy indoor air pollution, especially elevated of CO (Naeher et al., 2000). Yak dung is the primary energy source for cooking and heating by nomadic Tibetan herders (Li et al., 2012). Thus, factor 5 is attributed to the burning of yak dung for residential heating and cooking.

## 4. Conclusion

The QTP is an ideal location for characterizing aerosol properties. In this study, we investigated the characterizations of WSIs associated with autumn $PM_{2.5}$ at a background site (3295 m a.s.l.) in the QTP. Real time levels of WSIs, $PM_{2.5}$, gaseous pollutants, and meteorological parameters were collected to analyze the ion chemistry of aerosols in the QTP. $SO_4^{2-}$, $NO_3^-$, and $NH_4^+$ (SNA) were the three most abundant WSI species, and crust-originated ions ($Na^+$, $Mg^{2+}$, $K^+$, and $Ca^{2+}$) comprised a small fraction of total WSIs. As compared to similar studies in China, SNA concentrations in this study were lower as compared to low altitude urban areas, but higher relative to other sites in the QTP.

Further investigation regarding the formation of $SO_4^{2-}$ and $NO_3^-$ revealed that strong solar intensity and high $O_3$ concentrations combined with low daytime RH greatly enhanced the conversion of $SO_2$ and $NO_2$ to $SO_4^{2-}$ and $NO_3^-$, respectively. Heterogeneous reactions were weak overnight, and contributed to $SO_4^{2-}$ formation only. Our analysis suggests that photochemical reactions played a critical role in the formation of $SO_4^{2-}$ and $NO_3^-$ during our observation period.

Source apportionment using a PMF model identified five factors: salt lake emissions, mixed factor emissions (livestock feces, biomass burning, and crustal material emissions), traffic emissions, secondary inorganic aerosols, and residential burning. With the exception of some natural sources (salt lake, livestock and crustal materials), anthropogenic emissions demonstrated a marked contribution to





particulate loading in the area. For example, traffic emissions near the site, although
limited, still influenced local air quality. Hence, greater controls should be imposed
on diesel quality and heavy-duty truck emissions in this area to minimize traffic
pollution.

## Data Availability

All data of this work can be obtained from Bin Han (hanbin@craes.org.cn)

## Author Contributions

BH, WY and ZB designed the experiments. BY, XW and XD were in charge of the
whole field experiment. JW and XZ processed the original data and primary analysis.
BH prepared the manuscript with contributions from all co-authors

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
