# Peer review of "high-altitude area of China"

_Atmospheric Chemistry and Physics, 2018_

## Short Comment (SC1) · 7 Feb 2019

The paper is well written and covers a important topic relevant to air pollution in China. before being published I have the following comments and suggestions related to the manuscript: 1- " Researches" should be changed to "Research" because it is plural (lines 90, 159). You can also use "Studies".

Line 30: change understand to identify Line 152: include reference All downloaded

[Figure]

equations couldn't be checked (not clear) Line 216 add data after observations

Fig 4: Why high ozone peaks at hours 1 to 2 in Fig 4? Why does the ozone value at night increase to a value greater than that during the day?

Fig 4 SO4 displays wide fluctuation between hrs 12 and 20 with hour to hour fluctuations of about $2\mu$/m3, or about 15% of the absolute value. An explanation regarding the possible origins of these transients would be useful. Is this variance the true measure of the uncertainty of the SO4 ion concentration? If so, then the findings and conclusions should be recast in the context of the uncertainty of these data variabilities. If these data are averages over a number of days then correlation with wind direction etc. could be useful in identifying the origins of such variations and may aid in apportioning sources.

Fig 4 NO3 exhibits similar variability over the period of hour 12 to 18 with again wide fluctuations. Hours 14, 15 and 17 tend to be small while 16 is much higher. Again is this the true uncertainty of these data or are there other factors driving or pertaining to the variability (In both SO4 and NO3). A correlation plot with the prevailing wind direction and speed could be useful as these fluctuations may be a results of wind direction changes and may provide additional data regarding sources. In addition, these fluctuation will have significant influence on the NOR and SOR, at least an uncertainty value should be assigned to the NOR and SOR.

If the above points are considered in the context of the scatter of data displayed in figure 5. Since ozone and RH data trends shown in figure 4 appear to be relatively well defined, then the scatter in these data in figure 5 are consequently driven by the variability of the SOR and NOR. Following further, since SO2 and NO2 are also well defined in fig 4, then the variability in SOR and NOR is driven by the concentration of the respective WSI species. A careful analyses of this variability and a discussion with regards to the implications of these uncertainties to the conclusion drawn from figure 5 would be useful to the reader.

---

## Short Comment (SC2) · 19 Mar 2019

I would like to thank the Author to take the time to address my comments and I am satisfied with his answers and changes. Now, I think the publication of this paper that is well written and addressing major air pollution issues occurring in China is ready to be communicated to the world through an important Journal the "Atmospheric Chemistry and Physics" journal.

---

## Author Comment (AC1) · 19 Mar 2019

First, I would like to show my appreciation to Dr. Merched Azzi for his comments on this paper. I drafted my response as follows: The paper is well written and covers an important topic relevant to air pollution in China. Before being published I have the following comments and suggestions related to the manuscript:

1- " Researches" should be changed to "Research" because it is plural (lines 90, 159).

[Figure]

You can also use "Studies". Line 30: change understand to identify Line 152: include reference All downloaded equations couldn't be checked (not clear) Response: We will revised all of these above if we need to do further revision on this manuscript.

Line 216 add data after observations Response: All the data from our observation is shown in the first line of Table 2 Comparisons of WSIs concentrations with other high altitude and urban sites (mean, $\mu$g/m3). So if we need to do further revision, we will add "(as shown in Table 2)" after "observation".

Fig 4: Why high ozone peaks at hours 1 to 2 in Fig 4? Why does the ozone value at night increase to a value greater than that during the day? Response: To prove the diurnal change of ozone in our study, we referred to other publications. If we have chance to do further revision, we will add these two references and give a more detailed discussion on ozone diurnal variations. Wang et al. (2006) found ozone exhibited low levels in the morning and enhanced levels in the late afternoon and at nighttime during spring and summer time at Mount Waliguan (WLG, 36.28°N, 100.90°E, 3816 m above sea level), about 150 km south of our monitoring site. The diurnal trends of ozone in two seasons are shown in Fig.1. He explained that the lower morning values may be due to the depositional loss of ozone in stagnant air during the nighttime and early morning hours, and its enhancement in the afternoon could be due to the in situ photochemical production of ozone. Shen et al. (2014) monitored the ozone concentrations in Qinghai lake area (36°58′37′′N, 99°53′56′′E, and 3200 m above sea level), about 140km southwest of our monitoring site. As shown in Fig 2, a unimodal distribution was found for ozone concentrations in autumn time, with high levels in the afternoon and nighttime, and decreased level in the morning. The lower morning values might be due to the depositional losses in stagnant air during the nighttime and early morning hours, and its enhancement in the late afternoon and night could be due to downward transport of the free troposphere air.

Fig 4 SO4 displays wide fluctuation between hrs 12 and 20 with hour to hour fluctuations of about $2\mu$/m3, or about 15% of the absolute value. An explanation regarding the

possible origins of these transients would be useful. Is this variance the true measure of the uncertainty of the SO4 ion concentration? If so, then the findings and conclusions should be recast in the context of the uncertainty of these data variabilities. If these data are averages over a number of days then correlation with wind direction etc. could be useful in identifying the origins of such variations and may aid in apportioning sources. Fig 4 NO3 exhibits similar variability over the period of hour 12 to 18 with again wide fluctuations. Hours 14, 15 and 17 tend to be small while 16 is much higher. Again is this the true uncertainty of these data or are there other factors driving or pertaining to the variability (In both SO4 and NO3). A correlation plot with the prevailing wind direction and speed could be useful as these fluctuations may be a results of wind direction changes and may provide additional data regarding sources. In addition, these fluctuation will have significant influence on the NOR and SOR, at least an uncertainty value should be assigned to the NOR and SOR. If the above points are considered in the context of the scatter of data displayed in figure 5. Since ozone and RH data trends shown in figure 4 appear to be relatively well defined, then the scatter in these data in figure 5 are consequently driven by the variability of the SOR and NOR. Following further, since SO2 and NO2 are also well defined in fig 4, then the variability in SOR and NOR is driven by the concentration of the respective WSI species. A careful analyses of this variability and a discussion with regards to the implications of these uncertainties to the conclusion drawn from figure 5 would be useful to the reader. Response: We have noticed this problem. However, we believe it might be caused by some extremely high or low observations during the campaign. Also we did the correlation analysis between concentrations of sulfate or nitrate and wind direction or speed separately in each time period, and the correlation coefficients are shown in Table 1. The bold and colored figures are the questionable time period in Dr. Azzi's comments. Actually we cannot find any clue from this table. Therefore, our primary conclusion is that the existence outliers are believed to be the potential reason to cause the large variations of averaged sulfate and nitrate during afternoon and early evening. We will go to find the outliers and evaluate the reliability of these data. If not reliable, we will

delete them in our future revision. If reliable, we will add some explanations on how to interpret the large variations, and how it will influence the trend of SOR and NOR.

Reference: Shen, Z.X., Cao, J.J., Zhang, L.M., Zhao, Z.Z., Dong, J.G., Wang, L.Q., Wang, Q.Y., Li, G.H., Liu, S.X., Zhang, Q., 2014. Characteristics of surface O-3 over Qinghai Lake area in Northeast Tibetan Plateau, China. Science of the Total Environment 500, 295-301. Wang, T., Wong, H.L.A., Tang, J., Ding, A., Wu, W.S., Zhang, X.C., 2006. On the origin of surface ozone and reactive nitrogen observed at a remote mountain site in the northeastern Qinghai-Tibetan Plateau, western China. J. Geophys. Res.-Atmos. 111.

[Figure]

(a) spring    (b) summer

Fig. 1 Average diurnal patterns of O3, CO at WLG for (a) spring and(b) summer;
error bars indicate standard errors

**Fig. 1.**

Fig 2. Diurnal cycle of O3concentration in Qinghai Lake area

**Fig. 2.**

Table 1 Correlation coefficients between concentrations of sulfate or nitrate and wind direction or speed

| Time | Nitrate-Wind Direction | Nitrate-Wind Speed | Sulfate-Wind Direction | Sulfate-Wind Speed |
|---|---|---|---|---|
| 0:00 | 0.14 | -0.16 | -0.23 | 0.52 |
| 1:00 | 0.16 | 0.21 | -0.74 | 0.69 |
| 2:00 | 0.00 | 0.45 | -0.08 | 0.59 |
| 3:00 | -0.12 | 0.45 | -0.74 | 0.89 |
| 4:00 | 0.45 | 0.05 | -0.41 | 0.86 |
| 5:00 | 0.45 | -0.18 | -0.36 | 0.82 |
| 6:00 | -0.26 | 0.03 | 0.01 | 0.68 |
| 7:00 | 0.17 | 0.22 | -0.11 | 0.55 |
| 8:00 | -0.08 | 0.53 | -0.30 | -0.02 |
| 9:00 | -0.10 | 0.34 | 0.38 | 0.31 |
| 10:00 | -0.39 | 0.36 | -0.51 | -0.14 |
| 11:00 | **-0.03** | **0.22** | **-0.43** | **0.55** |
| 12:00 | **-0.63** | **0.35** | **-0.20** | **0.33** |
| 13:00 | **0.36** | **0.72** | **0.53** | **-0.16** |
| 14:00 | **-0.09** | **0.06** | **0.28** | **0.74** |
| 15:00 | **0.05** | **-0.07** | **-0.03** | **0.17** |
| 16:00 | **0.56** | **0.21** | **0.18** | **0.11** |
| 17:00 | **0.29** | **-0.28** | **0.37** | **0.05** |
| 18:00 | -0.51 | -0.35 | **0.73** | **0.56** |
| 19:00 | 0.15 | -0.12 | **-0.68** | **-0.07** |
| 20:00 | -0.50 | 0.07 | 0.34 | -0.57 |
| 21:00 | -0.07 | -0.04 | -0.29 | -0.59 |
| 22:00 | 0.34 | -0.31 | 0.19 | -0.05 |
| 23:00 | -0.32 | -0.02 | -0.50 | 0.74 |

**Fig. 3.**

---

## Referee Comment (RC1) · Anonymous Referee #1 · 9 Apr 2019

In this work, the authors did a field observation in the northeast of Tibetan Plateau, including the particulate matters as well as trace gases. Although such study is meaningful for the better understanding the atmospheric chemistry over this region, the current version of the manuscript suffer major problems.

Specific comments: 1. In the introduction parts, the authors should state the motivates of this study more clearly. Several papers have been published for this site (Menyuan)

in this special issue. Based on the previous studies, line 80-88, what the knowledge gaps or questions still exist for this region? 2. In the section of Methods (sampling site), the Qinghai lake (major source of sulfate found later of this study) and major traffic roads (major source of NOx in this study) should also be introduced. 3. In the section 2.2, why CI- data was missing? What's the data quality of this online monitoring? Did you compare it with the traditional method (filter sampling + IC)? What's the detection limits of the trace gases? 4. For the ion balance, without the CI- data, it is somewhat strange to see that anion is only composed of sulfate and nitrate. 5. Line 158-173, there is no need to describe the the basic theory of PMF in such detailed way 6. Regarding the contents in table 2, some locations may be unnecessary to include.Line 221, are you sure the study site of Kumar and Sarin (2010) is urban area? 7. I understand the NO3-/SO42- is frequently used to indicate the relative importance of vehicle and coal combustion. Such works were mostly based in urban or populated area like North China Plains or South Asia. However, such ratio seems not applicable for this study (Menyuan) for several reasons. First, as stated in Line 257 and later (by PMF results), biomass burning is also important source of nitrogen species. For sulfate, besides the coal combustion, salt lakes (like Qinghai Lake) were also proposed at important source of sulfate (see more details in PMF parts). 8. Line 267-269, actually, this is no data of concentration abundance of organic matters yet. So it is not so conniving to say the particle growth is caused by organics. 9. Some time, the authors say "particle growth" sometime, "PM2.5 increase" were used. So is there any difference between such two expression? particle growth means hydroscopic growth in terms of mass or size?? 10. Section 3.2, regarding the diurnal variations, what's the role of meteorologic factors like PBL? 11. Line 298, what's the meaning of remote transportation? 12. Line 308-311, maybe crustal materials is responsible for the increase of PM2.5 13. Line 332-334, it is strange to see such description here. You know such points actually were established after the discussion for Figure 6. 14. In the previous studies, the aerosols and rain over Tibetan Plateau were found to be alkaline. However, in this study, the aerosols were found to be acid. SO more discussion (more references for Tibetan
Plateau) is also expected, to make this point more convincing. 15. For the source apportionment by PMF, the ions were only ascribed to two factors. I.e., factor 1, salt lake and factor 2, mixed. it seems PMF failed to identify the sources of those inorganic ions adequately. 16. Currently, the writing of the conclusion part is very weak. What's the values and implications of this work for the international scientific communities? Compared with the statements in the Introduction parts, what's questions have been addressed after the study? 17. Line 487-488, the authors stated that "Our analysis suggests that photochemical reactions played a critical role in the formation of SO42- and NO3- during our observation period." However, salt lake emission was identified as the first factor (for SO42-) by PMF. Such expressions seem contradict.

---

## Referee Comment (RC2) · Anonymous Referee #2 · 23 May 2019

The manuscript covers an important topic and key area relevant to the background level air pollution in China. It gave an overall view of ion characterizations with PM2.5 at a remote site of the QTP. However, I have some concerns on this submission. About PMF model, I think source apportionment of PM2.5 should be based on the full dataset of chemical compositions of particles, including ions, carbons and elements. However, in this study, only ions data was included, while carbons and elements were not analyzed. This might not be in line with the principle of source apportionment. Another concern is

that the authors included gases pollutants data in the model, but they are not chemical compositions of PM. The authors discussed a lot the secondary formation of sulfate and nitrate; however, part of sulfate was thought to be emitted directly by salt lake in PMF result. The secondary formation and direct emission of sulfate seem to exist simultaneously, but the authors failed to explicit them in their analysis. I suggested the author delete the section of PMF to avoid contradiction between different sections of manuscript, or give more detailed explanation on them. High relationship between $SO_4^{2-}$ and $Na^+$ was found in this study, however, detailed discussion should be done in section 3.4. This is also important for the source analysis of sulfate.

---

## Author Comment (AC2) · 31 May 2019

We appreciate the comments from Anonymous Referee #1. His or her comments gave us a lot of chances to reevaluate our manuscript. We have made corresponding revision according to his comments. We attached our revised manuscript in Supplement. All our responses are as follows:

In this work, the authors did a field observation in the northeast of Tibetan Plateau, in-

[Figure]

cluding the particulate matters as well as trace gases. Although such study is meaningful for the better understanding the atmospheric chemistry over this region, the current version of the manuscript suffer major problems. Specific comments:

1. In the introduction parts, the authors should state the motivates of this study more clearly. Several papers have been published for this site (Menyuan) in this special issue. Based on the previous studies, line 80-88, what the knowledge gaps or questions still exist for this region? Response to reviewer: We added some sentences in line 88-97 to better express the gaps that still exist and the advantage by using high resolution monitoring equipment.

2. In the section of Methods (sampling site), the Qinghai lake (major source of sulfate found later of this study) and major traffic roads (major source of NOx in this study) should also be introduced. Response to reviewer: We added the introductions on the Qinghai Lake and surrounding major road in line 127-128 and 134-136.

3. In the section 2.2, why Cl- data was missing? What's the data quality of this on-line monitoring? Did you compare it with the traditional method (filter sampling + IC)? What's the detection limits of the trace gases? Response to reviewer: 1) Cl- data missing: The analysis of Cl- was greatly interfered during whole sampling period. There are some interference/background peaks near the retention time of Cl-. After background peak being removed, we got negative value of Cl-. Therefore, we believed there were some problems in the analysis of Cl- during this time. We chose to remove all the data of Cl-. 2) Comparison between URG and traditional filter sampling: Actually we didn't do any comparison sampling with filter during our campaign, and we should admit it is our problem on initial study design. 3) The detection limits of the trace gases: We added the limits of detection of equipment for trace gases in line 145-146.

4. For the ion balance, without the Cl- data, it is somewhat strange to see that anion is only composed of sulfate and nitrate. Response to reviewer: We totally understand the concern of reviewer. We should admit that it does seem strange without Cl- in

ion balance analysis. However, given that the analysis data of Cl- was not good and had been removed in our preliminary analysis, we can do nothing now on present ion balance discussion. In our future potential observation, we will try to solve this problem.

5. Line 158-173, there is no need to describe the basic theory of PMF in such detailed way Response to reviewer: We moved some parts of introduction on miss data treatment and choice on the number of the factors to supplementary material.

6. Regarding the contents in table 2, some locations may be unnecessary to include. Line 221, are you sure the study site of Kumar and Sarin (2010) is urban area? Response to reviewer: 1) We deleted some locations in the table 2. 2) We checked the study of Kumar and Sarin (2010) again, and it turned out we were wrong. But it is unnecessary to the Table 2 now, so we chose to delete it.

7. I understand the NO3-/SO42- is frequently used to indicate the relative importance of vehicle and coal combustion. Such works were mostly based in urban or populated area like North China Plains or South Asia. However, such ratio seems not applicable for this study (Menyuan) for several reasons. First, as stated in Line 257 and later (by PMF results), biomass burning is also important source of nitrogen species. For sulfate, besides the coal combustion, salt lakes (like Qinghai Lake) were also proposed at important source of sulfate (see more details in PMF parts).

Response to reviewer: We totally agree with reviewer's opinion on this point. The NO3-/SO42- is really more applicable in urban area. In the QTP where we studied, it is questionable to use it as a marker. So we deleted all the expressions on NO3-/SO42- to avoid some possible misinterpretations.

8. Line 267-269, actually, this is no data of concentration abundance of organic matters yet. So it is not so conniving to say the particle growth is caused by organics. Response to reviewer: It is true we don't have data to prove that, and can only refer to other studies. So we deleted the expressions on the organic matter, and added a sentence "Therefore, more observational campaign should be implemented in the future

to investigate the driver compositions on the increase of PM2.5 mass concentrations."

9. Some time, the authors say "particle growth" sometime, "PM2.5 increase" were used. So is there any difference between such two expression? particle growth means hydroscopic growth in terms of mass or size?? Response to reviewer: We changed all "particle growth" to "PM2.5 increase". When we said "particle growth" or "PM2.5 increase", we both meant the concentration increase of PM2.5. Obviously, "particle growth" doesn't mean this. Just like the reviewer said, it should mean hydroscopic growth in terms of mass or size. So we corrected it.

10. Section 3.2, regarding the diurnal variations, what's the role of meteorologic factors like PBL? Response to reviewer: We didn't do any observations on PBL, and it is our fault in the study design. We also asked some other collaborators to see if we can be accessible to the PBL data of Mount Waliguan. Unfortunately, no data could be available during our observation period in that site, either. We are not sure if the PBL data in Xining city is suitable here, given that the altitude difference between our site and Xining city is quite large. Therefore, we have to leave this analysis blank for further study.

11. Line 298, what's the meaning of remote transportation? Response to reviewer: It is supposed to mean the long-range transportation. To avoid confusion on the understanding of oxidation ratio, we deleted the sentence " Furthermore, remote transportation also contributed to the concentrations of SO42- and NO3-, thus increasing the oxidant ratio, despite not being products of local oxidation."

12. Line 308-311, maybe crustal materials is responsible for the increase of PM2.5 Response to reviewer: We showed the trends of the percentages of all WSI in PM2.5 in Figure 3, and none of them turned out to be responsible for the increase of PM2.5. So here we added a sentence " Crustal materials are either not responsible for it as shown in Figure 3." at line 338 to make it more clear.

13. Line 332-334, it is strange to see such description here. You know such points

actually were established after the discussion for Figure 6. Response to reviewer: We moved the description to the line 374-377.

14. In the previous studies, the aerosols and rain over Tibetan Plateau were found to be alkaline. However, in this study, the aerosols were found to be acid. So more discussion (more references for Tibetan Plateau) is also expected, to make this point more convincing. Response to reviewer: We summarize some papers published about the aerosol ions on the QTP and rewrote the discussion in this section. The results are found to be quite different at different locations. In the studies of South edge of the QTP, the aerosols were found to be alkaline at Qomolangma (Mt. Everest) Station for Atmospheric and Environmental Observation and Research Station (TSP, C/A=4.1) (Cong et al. 2015), four sites in central Himalayan region (TSP, C/A=3.7) (Tripathee et al. 2017), and Shigatsz, China (PM2.1, C/A=1.5) (Yang et al. 2016). While in the studies of the northeastern QTP, the results varied with locations. Two studies at Qilan Shan Station at different time achieved difference values of C/A (C/A=1.3, sampling time: summertime of 2011; C/A=0.95, sampling time: summertime of 2012) (Xu et al. 2014; Xu et al. 2015). Another study at the Qinghai Lake also got slightly acidic result (PM2.5, C/A=0.8) (Zhao et al. 2015). In their study at the summer of 2012, Xu et al. (2015) also found that the equivalent balances of water-soluble species in different size modes indicate that the accumulation mode particles were somewhat acidic (with the linear regression slope of [NH4++Ca2++Mg2++K+] vs. [SO42-+NO3-] being 0.6) and that the coarse mode particles were almost neutral (the slope was 0.999), indicating that small size of particles show tendency of acid.

15. For the source apportionment by PMF, the ions were only ascribed to two factors. I.e., factor 1, salt lake and factor 2, mixed. It seems PMF failed to identify the sources of those inorganic ions adequately. Response to reviewer: Integrating other comments of the reviewer, we changed the input of the PMF model and also rearranged the structure of the paper. First, we excluded all gases pollutants in the PMF model and allowed only ion data in the model. Thus we got five factors including animal waste emission

and biomass burning, crustal dust, salt lake emissions, secondary sulfate, secondary nitrate. Second, we moved the PMF model results to section 3.2 to use some of them (the results of secondary sulfate and nitrate) for the following SOR and NOR calculation.

16. Currently, the writing of the conclusion part is very weak. What's the values and implications of this work for the international scientific communities? Compared with the statements in the Introduction parts, what's questions have been addressed after the study? Response to reviewer: We rewrote part of the conclusion and added some implications of this work for the international scientific communities "To our knowledge, there is no such real-time measurement on WSIs associated with PM2.5 at rural sites in the QTP yet. This study provides some preliminary results on aerosol ion compositions on the QTP, and proposes the potential formation mechanism of secondary sulfate and nitrate. These findings are supposed to be useful for further studies on aerosol chemistry in this area." We also addressed some questions after the study by saying "In this study, we finished some analysis on WSIs by taking advantage of real-time data to: 1) analyze the diurnal variations of WSIs; 2) discuss the formation of secondary sulfate and nitrate at the QTP; and 3) investigate source apportionment on hourly data within short-term observation. All these above are difficult by using traditional manual PM2.5 sampling, given that it usually takes hours or even days for sample collection, and is unable to detect more variations on aerosol compositions and supply more data on finer temporal scale for further analysis. "

17. Line 487-488, the authors stated that "Our analysis suggests that photochemical reactions played a critical role in the formation of SO42- and NO3- during our observation period." However, salt lake emission was identified as the first factor (for SO42-) by PMF. Such expressions seem contradict. Response to reviewer: We added "After excluding the emission of sulfate from the salt lake" here. Also in the section of "results and discussion", we ran the PMF model again by using only ions data, and excluded the ratio of sulfate emitted from the salt lake. Thus we used sulfate and nitrate concentrations after modelling calculation for SOR and NOR discussion.

Reference:

Cong Z, Kang S, Kawamura K, Liu B, Wan X, Wang Z, et al. 2015. Carbonaceous aerosols on the south edge of the tibetan plateau: Concentrations, seasonality and sources. Atmos Chem Phys 15:1573-1584. Tripathee L, Kang SC, Rupakheti D, Cong ZY, Zhang QG, Huang J. 2017. Chemical characteristics of soluble aerosols over the central himalayas: Insights into spatiotemporal variations and sources. Environ Sci Pollut Res 24:24454-24472. Xu J, Wang Z, Yu G, Qin X, Ren J, Qin D. 2014. Characteristics of water soluble ionic species in fine particles from a high altitude site on the northern boundary of tibetan plateau: Mixture of mineral dust and anthropogenic aerosol. Atmospheric Research 143:43-56. Xu JZ, Zhang Q, Wang ZB, Yu GM, Ge XL, Qin X. 2015. Chemical composition and size distribution of summertime pm$_{2.5}$ at a high altitude remote location in the northeast of the qinghai–xizang (tibet) plateau: Insights into aerosol sources and processing in free troposphere. Atmos Chem Phys 15:5069-5081. Yang YJ, Zhou R, Yan Y, Yu Y, Liu JQ, Di YA, et al. 2016. Seasonal variations and size distributions of water-soluble ions of atmospheric particulate matter at shigatse, tibetan plateau. Chemosphere 145:560-567. Zhao ZZ, Cao JJ, Shen ZX, Huang RJ, Hu TF, Wang P, et al. 2015. Chemical composition of pm2.5 at a high-altitude regional background site over northeast of tibet plateau. Atmos Pollut Res 6:815-823.

Please also note the supplement to this comment:
https://www.atmos-chem-phys-discuss.net/acp-2018-1345/acp-2018-1345-AC2-supplement.pdf

---

## Author Comment (AC3) · 31 May 2019

We appreciate the comments from Anonymous Referee #2. His or her comments helped us a lot in improving our manuscript. We have made corresponding revision according to his comments. We attached our revised manuscript in Supplement. All our responses are as follows:

The manuscript covers an important topic and key area relevant to the background

[Figure]

level air pollution in China. It gave an overall view of ion characterizations with PM2.5 at a remote site of the QTP. However, I have some concerns on this submission.

About PMF model, I think source apportionment of PM2.5 should be based on the full dataset of chemical compositions of particles, including ions, carbons and elements. However, in this study, only ions data was included, while carbons and elements were not analyzed. This might not be in line with the principle of source apportionment. Response to reviewer: I agree with the reviewer's opinion on the application of the PMF model. However, traditional application of the PMF model is based on the source apportionment of the mass of PM2.5, therefore, it requires the full dataset of PM2.5 compositions. However, in our study, we used the PMF model only for distinguish the potential sources of water-soluble inorganic ions, not the PM2.5 mass. So only the ions data is enough for the model. So previous studies also used only ions data for source apportionment according to their purpose (Han et al., 2016; Shi et al., 2017).

Another concern is that the authors included gases pollutants data in the model, but they are not chemical compositions of PM. Response to reviewer: We agree with the reviewer's point. So we excluded all gases pollutants in the PMF model and allowed only ion data in the model. Thus we got five factors including animal waste emission and biomass burning, crustal dust, salt lake emissions, secondary sulfate, secondary nitrate

The authors discussed a lot the secondary formation of sulfate and nitrate; however, part of sulfate was thought to be emitted directly by salt lake in PMF result. The secondary formation and direct emission of sulfate seem to exist simultaneously, but the authors failed to explicit them in their analysis. I suggested the author delete the section of PMF to avoid contradiction between different sections of manuscript, or give more detailed explanation on them. Response to reviewer: According to the comments from both reviewers, we ran the PMF model again by using only ions data, and excluded the ratio of sulfate emitted from the salt lake. Thus we used sulfate and nitrate concentrations after modeling calculation for SOR and NOR discussion. The method was

Interactive
comment

shown at line 198-204. Also in the conclusion, we added "After excluding the emission of sulfate from the salt lake" at line 482.

High relationship between SO42- and Na+ was found in this study, however, detailed discussion should be done in section 3.4. This is also important for the source analysis of sulfate. Response to reviewer: We added some analysis at line 290-292, which reads "Factor 3 has high Na+ loading and a moderate SO42- loading, which is also shown in previous correlation analysis with the correlation coefficient between SO42- and Na+ is 0.76."

Reference:

Han, B., Zhang, R., Yang, W., Bai, Z. P., Ma, Z. Q., and Zhang, W. J.: Heavy haze episodes in Beijing during January 2013: Inorganic ion chemistry and source analysis using highly time-resolved measurements from an urban site, Science of the Total Environment, 544, 319-329, 10.1016/j.scitotenv.2015.10.053, 2016. Shi, G. L., Xu, J., Peng, X., Xiao, Z. M., Chen, K., Tian, Y. Z., Guan, X. B., Feng, Y. C., Yu, H. F., Nenes, A., and Russell, A. G.: pH of Aerosols in a Polluted Atmosphere: Source Contributions to Highly Acidic Aerosol, Environmental Science & Technology, 51, 4289-4296, 10.1021/acs.est.6b05736, 2017.

Please also note the supplement to this comment:
https://www.atmos-chem-phys-discuss.net/acp-2018-1345/acp-2018-1345-AC3-supplement.pdf
* * *
[Figure]

**Supplement:**

[revised manuscript text omitted]

Figure 1 Location of sampling site

**2.2 Instruments**

Hourly concentrations of $NO_3^-$, $SO_4^{2-}$, $Na^+$, $NH_4^+$, $K^+$, $Mg^{2+}$, and $Ca^{2+}$ associated with $PM_{2.5}$ were simultaneously measured by an ambient ion monitor (Model URG 9000B, URG Corporation, USA). A set of commercial instruments from Teledyne API (USA) were equipped to measure hourly concentrations of $SO_2$ (M100EU, detection limit: 50 ppt), $NO/NO_2/NO_x$ (M200EU, detection limit: 50 ppt), CO (Model 300EU, detection limit: 20 ppb), and $O_3$ (Model 400E, detection limit: 0.6 ppb). Hourly $PM_{2.5}$ concentrations were measured using an Ambient Dust Monitor 365

(GRIMM; Grimm Aerosol Technik GmbH &Co. KG, Ainring, Germany).
Meteorological parameters (e.g. temperature, relative humidity, pressure, and wind
speed and direction) were also recorded.
**153 2.3 data analysis**
2.3.1 Oxidant ratio
Particulate sulfate and nitrate oxidation ratios (SOR and NOR, respectively),
defined as the molar ratio of $SO_4^{2-}$ and $NO_3^-$ to total oxidized sulfur and nitrogen
(Zhou et al., 2009), were used to evaluate secondary conversion from $NO_2$ and $SO_2$ to
$NO_3^-$ and $SO_4^{2-}$, respectively. High SOR and NOR indicate larger conversions of $SO_2$
and $NO_x$ to their respective particulate forms in $PM_{2.5}$. In this study, NOR and SOR
were calculated based on the following formulae:

$$SOR = \frac{[SO_4^{2-}]}{[SO_2]+[SO_4^{2-}]} \quad (1)$$

$$NOR = \frac{[NO_3^-]}{[NO_2]+[NO_3^-]} \quad (2)$$

Given that there potentially exist the direct emissions of sulfate and nitrate (such
as the sulfate released from the salt lake, which will be discussed in the source
apportionment section), it is not suitable to apply their concentrations directly into the
formulae (1) and (2). Therefore, we used the PMF model to calculate the
concentrations of secondary sulfate and nitrate in each observation, and then take
these modeled concentrations to estimate the SOR and NOR. The PMF model will be
introduced in following section.
2.3.2 Ion balance
Ion balance was used to evaluate the acid-base balance of aerosol particles. We
converted the WSIs mass concentration into an equivalent concentration, as follows:

$$C\ (cation, \mu eq/m^3) = \frac{Na^+}{23} + \frac{NH_4^+}{18} + \frac{K^+}{39} + \frac{Mg^{2+}}{12} + \frac{Ca^{2+}}{20} \quad (3)$$

$$A\ (anion, \mu eq/m^3) = \frac{SO_4^{2-}}{48} + \frac{NO_3^-}{62} \quad (4)$$

2.3.3 Source apportionment
Positive Matrix Factorization (PMF), developed by Paatero (Paatero and Tapper,
1994; Paatero, 1997), has been widely applied in source apportionment researches. In
this model, a data matrix $X_{ij}$, in which $i$ is the sample and $j$ is the measured chemical
species, can be viewed as a speciated data set, and the concept of this model can be
represented as:

$$X_{ij} = \sum_{k=1}^{p} g_{ik} f_{kj} + e_{ij} \quad (5)$$

where $p$ is the number of factors; $f$ is the chemical profile of each source, $g$ is
the mass contribution of each factor to the sample; $f_{jk}$ is the source profile, and $e_{ij}$
is the residual for each species or sample.
PMF solves Eq (5) by minimizing the sum of the square of residuals weighted
inversely with the error estimates of the data points, Q, defined as:

$$Q = \sum_{i=1}^{n} \sum_{j=1}^{m} \left[ \frac{x_{ij} - \sum_{k=1}^{p} g_{ik} f_{kj}}{u_{ij}} \right]^2 \quad (6)$$

where $u_{ij}$ is the uncertainty of chemical species $j$ in sample $i$.

Furthermore, we used the results of the source apportionment to estimate the concentrations of secondary sulfate and nitrate and then applied them in the calculations of SOR and NOR. By running the PMF model, we identified the secondary sulfate and nitrate in factor $s$ and $t$, respectively. We also obtained the hourly contributions of each factor to the total ions, as well as the profile of each factor. Then the products of hourly contributions of factor $s$ or $t$ and the percentages of sulfate or nitrate in the profile of factor $s$ or $t$ are taken as the concentrations of secondary sulfate or nitrate. The calculation is defined as:

Presume that factor s and t are the secondary sulfate and nitrate sources calculated by the PMF model ($s, t \leq p$), then:

$$Contribution_{i,secondary\ sulfate} = g_{i,s} \times r_{s,sulfate} \quad (7)$$

$$Contribution_{i,secondary\ nitrate} = g_{i,t} \times r_{t\ nitrate} \quad (8)$$

where $g_{i,s}$ is the contribution of factor $s$ to sample $i$, while $g_{i,t}$ is the contribution of factor $t$ to sample $i$; and $r_{s,sulfate}$ is the ratio of $sulfate$ in the factor $s$, while $r_{t,nitrate}$ is the ratio of $nitrate$ in the factor $t$.

2.3.4 Statistical analysis

[revised manuscript text omitted]

To further examine the relationship between $PM_{2.5}$ and WSIs, we divided the $PM_{2.5}$ concentrations into four categories: a) $C(PM_{2.5}) < 20\mu g/m^3$, (b) $20\mu g/m^3 \leq C(PM_{2.5}) < 30\mu g/m^3$, (c) $30\mu g/m^3 \leq C(PM_{2.5}) < 40\mu g/m^3$, and (d) $C(PM_{2.5}) \geq 40\mu g/m^3$ and attributed each WSI measurement to its corresponding $PM_{2.5}$ category. Figure 3

shows the mean proportions of WSIs in PM$_{2.5}$ for the different categories. As the PM$_{2.5}$ concentration increases, the percentages of WSIs in PM$_{2.5}$ mass exhibited decreasing trends, suggesting that the contribution of WSIs to PM$_{2.5}$ increases was negligible. Therefore, more observational campaign should be implemented in the future to investigate the driver compositions on the increase of PM$_{2.5}$ mass concentrations.

[Figure]

Figure 3 Mass portions of WSIs within different PM$_{2.5}$ level ranges

**3.2 Source apportionment by PMF**

In this study, all WSIs and gaseous pollutants were introduced into the PMF model for source identification. Five factors were used in the PMF model. The distributions of the factor species and the percentage of total species are shown in Figure 4.

[Figure]

(a)                                        (b)

(c)                                        (d)

[Figure]

                                                      (e)

                    Figure 4 Source profiles exacted by PMF

(a) factor 1: animal waste emission and biomass burning; (b) factor 2: crustal dust; (c) factor 3:

salt lake emissions; (d) factor 4: secondary sulfate and (e) factor 5: secondary nitrate (blue circle represents the percentage of species sum, while white bar represents concentration of species)

High $NH_4^+$ and $K^+$ loading were observed in factor 1. Livestock feces, which is commonly found in the meadows around the sampling site is a possible source of $NH_4^+$. $K^+$ can be attributed to the combustion of biomass, which has both natural and anthropogenic origins. Li et al. (2015) found that occasional biomass burning events in the area contributed significantly to the formation of anthropogenic fine particles.. Consequently, factor 1 is attributed to mixed sources, animal emissions and biomass burning, and crustal materials. Factor 2 is identified as crustal materials with high loading of $Mg^{2+}$ and $Ca^{2+}$ (Ma et al., 2003). Factor 3 has high $Na^+$ loading and a moderate $SO_4^{2-}$ loading, which is also shown in previous correlation analysis with the correlation coefficient between $SO_4^{2-}$ and $Na^+$ is 0.76. Our monitoring site is approximately 100 km from Qinghai Lake, a saline and alkaline water body which is the largest lake in China. Zhang et al. (2014) collected total suspended particle (TSP) and $PM_{2.5}$ samples at Qinghai Lake, and found that the concentration of $Na^+$ was higher as compared to other mountainous areas. Furthermore, they found that $SO_4^{2-}$ was one the most abundant species in both TSP and $PM_{2.5}$. Therefore, we attribute factor 1 to aerosols emitted from Qinghai Lake. Factor 4 and 5 are enriched with $SO_4^{2-}$ and $NO_3^-$, respectively, which could be considered as the secondary sulfate and secondary nitrate. The precursor of $SO_4^{2-}$ is $SO_2$, which may originate from coal combustion, and $NO_3^-$ is mainly converted from ambient $NO_x$, emitted by both vehicle exhaust and fossil fuel combustion.

**3.2 Diurnal variation analysis**

[revised manuscript text omitted]

Figure 8 Scatter plot of (a) $[NH_4^+]$ and $[NO_3^-]$ (b) $[Na^+]$ and $[SO_4^{2-}]$ (c) [excessive $NH_4^+$] and [excessive $SO_4^{2-}$] (d) $[NH_4^+ + Na^+ + K^+]$ and $[NO_3^- + SO_4^{2-}]$

**3.5 Ion acidity analysis**

The ion balance, expressed by the sum of the equivalent concentration ($\mu eq/m^3$) ratio of cation to anion (C/A), is an indicator of the acidity of particulate matter (Wang et al., 2005). In this study, the ion balance ratio was 0.87, indicating that aerosols tended to be acid, in line with previous studies in the QTP (Xu et al., 2015; Zhao et al., 2015). However, the results across the QTP are quite different at different locations. In the studies of south edge of the QTP, the aerosols were found to be alkaline at Qomolangma (Mt. Everest) Station for Atmospheric and Environmental Observation and Research Station (TSP, C/A=4.1) (Cong et al., 2015), four sites in central Himalayan region (TSP, C/A=3.7) (Tripathee et al., 2017), and Shigatsz, China (PM2.1, C/A=1.5) (Yang et al., 2016). While in the studies of the northeastern QTP, the results vary. Two studies at Qilan Shan Station (QSS) at different time achieved difference values of C/A (C/A=1.3, sampling time: summer, 2011; C/A=0.95, sampling time: summer, 2012) (Xu et al., 2014; Xu et al., 2015). Another study at the Qinghai Lake also got slightly acidic result (PM$_{2.5}$, C/A=0.8) (Zhao et al.,

2015). In their study at the summer of 2012, Xu et al. (2015) also found that the
equivalent balances of water-soluble species in different size modes indicate that the
accumulation mode particles were somewhat acidic (with the linear regression slope
of $[NH4^{+}+Ca^{2+}+Mg^{2+}+K^{+}]$ vs. $[SO_4^{2-}+NO_3^{-}]$ being 0.6) and that the coarse mode
particles were almost neutral (the slope was 0.999), indicating that small size of
particles show tendency of acid. As compared to the results at the south edge of the
QTP that is mostly influenced by natural emission (such as mineral dust), the
northeastern QTP suffers more anthropogenic emissions (e.g. $SO_4^{2-}$ and $NO_3^{-}$), since it
is more close to the areas with intensive human activities.

Figure 9 shows the scatter and linear regression plot of cations and anions
($\mu eq/m^3$). It is apparent that most points are below the 1:1 line, highlighting the acid
tendency. The total equivalent anion concentration was regressed against the total
equivalent concentrations of cations, and the slope of regression was 0.58.

[Figure]

      Figure 9 Cation and Anion scatter plot and linear regression

Aerosol acidity for different categories of PM$_{2.5}$ concentration (described in
section 3.1) and their respective scatter plots of total equivalent concentrations of
anions and cations are shown in Figure 10. C/A was high when PM$_{2.5}$ concentrations
exceeded 20$\mu g/m^3$, indicating that aerosol acidity was weak when the PM$_{2.5}$
concentration was high. This provides further evidence to support our finding that
$SO_4^{2-}$ and $NO_3^{-}$ did not contribute to PM$_{2.5}$ increases.

[Figure]

Figure 10 Scatter plot Σ Anion and Σ Cation with different $PM_{2.5}$ concentration range (a) $C(PM_{2.5})$ < 20μg/m³ (b) 20μg/m³ ≤ $C(PM_{2.5})$ <30μg/m³ (c) 30μg/m³ ≤ $C(PM_{2.5})$ <40μg/m³ (d) $C(PM_{2.5})$ ≥ 40μg/m³

**4. Conclusion**

The QTP is an ideal location for characterizing aerosol properties. In this study, we investigated the characterizations of WSIs associated with autumn $PM_{2.5}$ at a background site (3295 m a.s.l.) in the QTP. In this study, we finished some analysis on WSIs by taking advantage of real-time data to: 1) analyze the diurnal variations of WSIs; 2) discuss the formation of secondary sulfate and nitrate at the QTP; and 3) investigate source apportionment on hourly data within short-term observation. All these above are difficult by using traditional manual PM2.5 sampling, given that it usually takes hours or even days for sample collection, and is unable to detect more variations on aerosol compositions and supply more data on finer temporal scale for further analysis.

During our observation, we collected real time concentrations of WSIs, and analyzed them together with $PM_{2.5}$, gaseous pollutants, and meteorological parameters for investigating ion chemistry of aerosols in the QTP. $SO_4^{2-}$, $NO_3^-$, and $NH_4^+$ (SNA) were the three most abundant WSI species, and crust-originated ions ($Na^+$, $Mg^{2+}$, $K^+$, and $Ca^{2+}$) comprised a small fraction of total WSIs. As compared to similar studies in China, SNA concentrations in this study were lower as compared to low altitude urban areas, but higher relative to other sites in the QTP. $NH_4NO_3$, $(NH_4)_2SO_4$, $Na_2SO_4$, and $K_2SO_4$ are found to be the major atmospheric aerosol components during our observation campaign.

Source apportionment using a PMF model identified five factors: mixed factor including animal waste emission and biomass burning, crustal dust, salt lake emissions, secondary sulfate and secondary nitrate. Based on the results of source apportionment, we found that the major sources of sulfate are salt lake emission and secondary transformation, while particulate nitrate is mostly from secondary conversion. After excluding the emission of sulfate from the salt lake, we investigated the possible formation pathway of $SO_4^{2-}$ and $NO_3^-$, the concentrations of which showed evident diurnal variations. The results revealed that strong solar intensity and high $O_3$ concentrations combined with low daytime RH greatly enhanced the conversion of $SO_2$ and $NO_2$ to $SO_4^{2-}$ and $NO_3^-$, respectively. Heterogeneous reactions were weak overnight, and contributed to $SO_4^{2-}$ formation only. Our analysis suggests that photochemical reactions played a critical role in the secondary formation of $SO_4^{2-}$ and $NO_3^-$ during our observation period.

To our knowledge, there is no such real-time measurement on WSIs associated with $PM_{2.5}$ at rural sites in the QTP yet. This study provides some preliminary results on aerosol ion compositions on the QTP, and proposes the potential formation mechanism of secondary sulfate and nitrate. These findings are supposed to be useful for further studies on aerosol chemistry in this area.

[revised manuscript text omitted]